# Boosting In-Context Learning in LLMs Through the Lens of Classical Supervised Learning

## Abstract

In-Context Learning (ICL) allows Large Language Models (LLM) to adapt to new tasks with just a few examples, but their predictions often suffer from systematic biases, leading to unstable performances in classification. While calibration techniques are proposed to mitigate these biases, we show that, in the logit space, many of these methods are equivalent to merely shifting the LLM's decision boundary without having the ability to alter its orientation. This proves inadequate when biases cause the LLM to be severely misdirected. To address these limitations and provide a unifying framework, we propose Supervised Calibration (SC), a loss-minimization based framework, which learns an optimal, per-class affine transformation of LLM's predictive probabilities in the logit space without requiring external data beyond the context. By using a more expressive functional class, SC not only subsumes many existing calibration methods in ICL as special cases but also enables the ability of altering and even completely reversing the orientation of the LLM's decision boundary. Furthermore, SC's loss-based nature facilitates the seamless integration of two purpose-built regularization techniques, context-invariance and directional trust-region regularizers. The former is designed to tackle the instability issue in ICL, while the latter is to control the degree of calibration. Finally, SC delivers state-of-the-art performance over calibration baselines in the 4-shot, 8-shot, and 16-shot settings across all nine datasets for Mistral-7B-Instruct-v0.3, Llama-2-7B-chat, and Qwen2-7B-Instruct.

## 1 Introduction

State-of-the-art LLMs exhibit a striking *in-context learning* (ICL) capability: with only a handful of input–label exemplars, they generalize to unseen queries almost as if they had been fine-tuned, thus functioning as highly sample-efficient few-shot learners (Brown et al., 2020; Liu and et al., 2023). However, a growing body of evidence shows that ICL performance can be brittle with respect to seemingly innocent design choices such as template wording (Min et al., 2022), verbaliser selection (Holtzman et al., 2021a), and the particular demonstrations given (Liu et al., 2022a). These biases and sensitivity of ICL pose a practical barrier to developing applications that are both adaptable and robust. Motivated by this, extensive research has been conducted to develop calibration approaches to address such a challenge for classification problems in ICL. The majority of calibration methods fall under label-marginal-based calibration (LM). These methods first estimate the LLM's probability for each label given the context alone via various approaches. They then discount the predictive probabilities of the LLM for the labels that are over-represented and boost those that are under-represented. See detailed discussion in the later sections.

Despite the empirical success of these methods, their ability of correcting the predictive probabilities of the LLM via its internal estimated prior is limited. Specifically, we show in Section 3.4 that the underlying idea of these methods is equivalent to optimally shifting the decision threshold of the base LLM. Hence, they are inherently incapable of altering or reversing the orientation of the decision boundary. This becomes

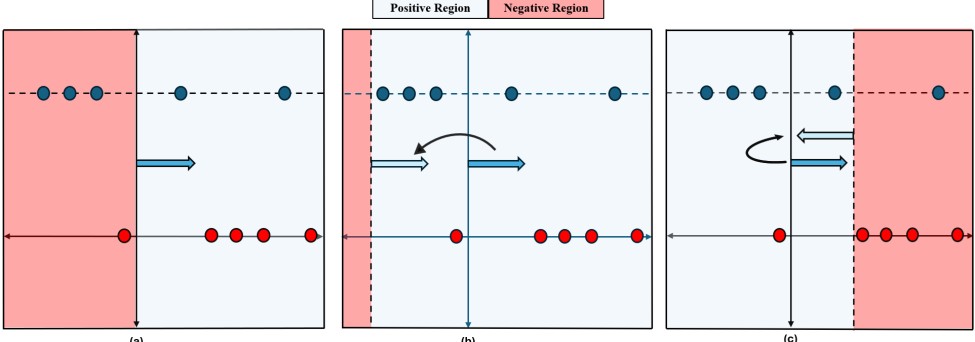

Figure 1: Comparison of ICL prediction strategies, where the x-axis represents the LLM's raw logits (log-odds). **(a)** *Base LLM (accuracy: 30%)*: The model predicts class 1 when logit $> 0$. **(b)** *Label Marginal Calibration (accuracy: 50%)*: These methods only shift the decision boundary, limiting correction when base LLM is systematically wrong. **(c)** *Supervised Calibration (accuracy: 80%)*: SC can shift and flip the decision boundary of the base LLM, resulting in a significant improvement.

problematic when the base LLM performs poorly. To further illustrate this limitation, consider a binary classification problem in Figure 1 (a), where the base LLM only achieves 30% accuracy. Since LM methods can only shift the decision threshold, their maximum improvement over the base LLM is capped, only achieving the level of random guessing as seen in Figure 1 (b). One may expect that such an issue becomes more common and severe in the multiclass classification, where distinguishing among a larger number of labels is inherently more difficult. For instance, on the SST-5 dataset, the average accuracy across three representative LLMs is only 22%, highlighting the severity of this challenge. This limitation motivates the need for a more principled calibration framework that is capable of correcting severely misaligned LLM predictions when necessary (e.g., by reversing the decision direction), and that subsumes existing methods as special cases while remaining both theoretically grounded and practically robust.

To achieve this goal, we introduce Supervised Calibration (SC), which is motivated by conceptualizing existing approaches as learning a calibrated classifier: they take a LLM's logits as input features and subsequently optimize a bias term to shift these logits. However, this shift only corresponds to moving the LLM's decision boundary to maximize the predictive accuracy illustrated in Figure 1 (b). Therefore, to enable more comprehensive adjustments, specifically, the ability to alter or reverse the orientation of the LLM's decision boundary, the proposed SC leverages the paradigm of loss-function-based classification and optimize both the bias and the scaling factor jointly. Our approach begins by generating a surrogate dataset, removing the necessity of external dataset beyond the given context. From this surrogate data, we extract features in the form of logits derived from the base LLM's output probabilities. Then we employ these features, paired with their corresponding true labels to train a standard classifier, which learns not only an optimal bias term but also an optimal rescaling factor. Critically, the concurrent optimization of this rescaling factor empowers our approach to reverse the LLM's decision boundary when advantageous (as illustrated in Figure 1 (c)). Moreover, the loss-minimization framework underpinning SC inherently supports the integration of regularization techniques designed for addressing the common problems in ICL and calibration. In this context, we propose a novel context-invariance regularizer for addressing the instability issue in ICL and a directional trust-region regularizer for controlling the degree of calibration. From a statistical viewpoint, these characteristics allow SC to pursue a balance regarding to the bias-variance trade-off. While SC's flexibility targets a reduction in approximation error over LM methods, its regularization components actively constrain variance which is an essential consideration within the data-scarce ICL paradigm. Collectively,

SC delivers an adaptable, stable, and theoretically grounded framework that improves LLMs' classification quality in few-shot settings, enabling fairer and more socially impactful applications as a result. Experimental results demonstrate that SC consistently outperforms existing calibration methods across a broad range of tasks, significantly enhancing the predictive performance of three distinct LLMs evaluated on nine inference datasets. For example, the performance of SC is striking on the SST-5 dataset with the Qwen model (8-shot setting), where it significantly outperforms baseline methods with accuracy from 25% (baselines) to 44%. This notable boost is directly attributable to its learned negative scaling factor which re-orients the base LLM decision boundary in this multiclass classification task. See Figure 4 for more details.

Our main contributions are summarized as follows: Firstly, we propose Supervised Calibration, which adopts loss minimization framework from classical supervised learning and calibrates ICL via learning optimal bias and scaling factors, enabling not only shifting but also altering the orientation of the base LLM decision boundary; Secondly, we integrate the context-invariance and directional trust region regularizations in SC, enhancing the stability of ICL and controlling the degree of the calibration respectively; Thirdly, we provide a theoretical intuition behind SC and its generalization over the LM methods; Lastly, we conduct extensive empirical studies to demonstrate the state-of-the-art performance of SC over several existing baselines.[1]

## 2 RELATED WORK

**Diagnosing biases and calibration via Label Marginal**. A seminal study by Zhao et al. (2021) identified primary in-context learning (ICL) biases—including majority-label, recency, and common-token bias—and introduced **Contextual Calibration (CC)**, which adjusts probabilities by normalizing against content-free prompts. Subsequently, observing that competition for probability mass degrades performance, Holtzman et al. (2021b) proposed **DCPMI** to recalibrate logits. Recent work has uncovered further ICL instabilities, such as feature and positional biases, with each diagnosis often paired with a lightweight calibration strategy (Si et al., 2023; Wang et al., 2023; Pezeshkpour and Hruschka, 2023). For instance, **Domain-Context Calibration (DC)** corrects predictions by averaging over random in-domain strings (Fei et al., 2023), while the more recent **Batch Calibration (BC)** uses unlabeled mini-batches to adjust each prediction (Zhou et al., 2023). Although these methods show empirical improvements, they can fail when the base LLM is substantially misaligned with the downstream task, as they cannot alter the model's decision direction. This limitation motivates the exploration of calibration frameworks with greater flexibility.

## 3 SUPERVISED CALIBRATION

### 3.1 BACKGROUND

Consider an $n$-class classification task with label verbaliser set $\mathcal{Y} = \{y_0, \ldots, y_{n-1}\}$ and query space $\mathcal{X}$. In few-shot in-context learning (ICL), the context $C_k$ is constructed by concatenating $k$ input–label exemplars $(x^{(i)}, y^{(i)})$ formatted via a template function $T$ such that $C_k = \mathrm{Concat}(T(x^{(1)}, y^{(1)}), \ldots, T(x^{(k)}, y^{(k)}))$. Then given the context of $k$-shots and a testing query $x \in \mathcal{X}$, the LLM predicts a label via computing $\hat{y} \in \arg\max_{y \in \mathcal{Y}} P_{\mathrm{LLM}}(y \mid x, C_k)$. While ICL offers an appealing alternative to the gradient-based fine-tuning by allowing LLMs to adapt to new tasks via only a handful of in-prompt demonstrations, the resulting posterior distribution $P_{\mathrm{LLM}}(y \mid x, C_k)$ is often distorted by some systematic biases. Such biases inherent in ICL often stems from context examples or their order, which makes $P_{\mathrm{LLM}}(y \mid x, C_k)$ significantly diverge from ground-truth posterior $P^*(y|x)$. Therefore, the objective of calibration is to refine LLM's predictive probabilities $P_{\mathrm{LLM}}(\cdot \mid x, C_k)$ to align with $P^*(y|x)$.

---

[1]Anonymized code for reproducibility: https://anonymous.4open.science/r/ICL-5CF5

Existing approaches are mainly focused on correcting the prior distribution of the label via estimating the LLM's internal prior given the context. Despite their successes, one can show that these approaches boils down to merely shifting the LLM's decision boundary, lacking the ability to alter an LLM's orientation. This limitation turns out to be essential especially in multi-class classification, where an LLM can easily make persistent mistakes. See Figure 1. Therefore, to further reduce the biases and align with $P^*(y \mid x)$ in such cases of substantial misorientation, we develop a more principle calibration called Supervised Calibration.

## 3.2 OUR PROPOSAL

To begin with, we assume the $k$ context examples $(x^{(i)}, y^{(i)})_{i=1}^k \overset{\text{i.i.d.}}{\sim} P^*$. Due to the aforementioned biases, the LLM's posterior $P_{\text{LLM}}(y \mid x, C_k)$ can deviate notably from the truth $P^*(y \mid x)$. In particular, we measure their deviation via the Kullback–Leibler (KL) divergence defined as $\mathbb{E}_{x \sim P^*} \left[ D_{\text{KL}} \big( P^*(\cdot \mid x) \,\|\, P_{\text{LLM}}(\cdot \mid x, C_k) \big) \right]$, where $D_{\text{KL}} \big( P \,\|\, Q \big) = \sum_{y \in \mathcal{Y}} P(y) \log \frac{P(y)}{Q(y)}$ for some probability measures $P$ and $Q$. Let $\Delta^n$ be the probability simplex over $\mathcal{Y}$. Then to correct for this, we seek a vector-valued calibration function $f^* : \Delta^n \to \Delta^n$, chosen from a prescribed class $\mathcal{F}$, such that when applied to the vector of LLM's predictive probabilities, it minimizes the KL-divergence, i.e.,

$$f^* = \arg\min_{f \in \mathcal{F}} \; \mathbb{E}_{x \sim P^*} [D_{\text{KL}}(P^*(\cdot \mid x) \,\|\, f(P_{\text{LLM}}(\cdot | x, C_k)))] = \arg\min_{f \in \mathcal{F}} \; -\mathbb{E}_{(x,y) \sim P^*}[\log(f_y(P_{\text{LLM}}(\cdot | x, C_k)))], \quad (1)$$

where $f_y$ is the $y^{th}$-coordinate projection of $f$. Note that as long as $\mathcal{F}$ contains the identity map, applying $f^*$ enhances the fidelity of $P_{\text{LLM}}$. To find $f^*$, we highlight two key challenges. Firstly, since our method is post-hoc, choosing an effective $\mathcal{F}$ operating solely on the base LLM predictive probabilities is essential. Secondly, there is no external data sampled from $P^*$ to approximate the objective function in Equation (1).

### 3.2.1 AFFINE-LOGIT APPROXIMATION AND LEAVE-SUBSET-OUT STRATEGY

To select an appropriate function class $\mathcal{F}$, we only need to consider $f$ defined over the log-odds of the predictive probabilities against a reference group (class 0 in this paper), since the logistic function is bijective. Specifically, denote the logits given by the base LLM as $\mathbf{m}(x; C_k) = \left( m_c(x; C_k) \triangleq \log \frac{P_{\text{LLM}}(y=c|x,C_k)}{P_{\text{LLM}}(y=0|x,C_k)} \right)_{c=1}^{n-1}$. Then, instead, we aim to choose the transformed function class $\widetilde{\mathcal{F}} = \big\{ f : \mathbb{R}^{n-1} \to \Delta^n \big\}$ for calibration. To facilitate it, notice that

$$P^*(y \mid x) = \frac{P^*(x \mid y) P^*(y)}{P^*(x)} \propto P_{\text{LLM}}(y \mid x, C_k) \frac{P^*(x \mid y)}{P_{\text{LLM}}(x \mid y, C_k)} \frac{P^*(y)}{P_{\text{LLM}}(y \mid C_k)} \quad (2)$$

$$\triangleq P_{\text{LLM}}(y \mid x, C_k) \, h(x, y, C_k), \quad (3)$$

which implies that

$$L_c^*(x) = m_c(x; C_k) + \underbrace{\log \left( \frac{P^*(x|c) P_{\text{LLM}}(x|0, C_k)}{P^*(x|0) P_{\text{LLM}}(x|c, C_k)} \right)}_{\text{Class Conditional Shift}} + \underbrace{\log \left( \frac{P^*(c) P_{\text{LLM}}(0|C_k)}{P^*(0) P_{\text{LLM}}(c|C_k)} \right)}_{\text{Label Marginal Shift}}, \quad (4)$$

$$\underbrace{\qquad\qquad\qquad\qquad\qquad\qquad\qquad\qquad\qquad\qquad\qquad\qquad}_{\log(h(x,c,C_k)/h(x,0,C_k))}$$

where $L_c^*(x) = \log(P^*(c|x)/P^*(0|x))$ is the true logit for class $c$. Thus, the primary challenge of choosing $\mathcal{F}$ lies in approximating the unknown correction term $\log(h(x, c, C_k)/h(x, 0, C_k))$. Since we only have access to the LLM's output logits $\mathbf{m}(x; C_k)$, we propose to approximate $\{L_c^*(x)\}_{c=1}^{n-1}$ via an affine transformation of $\{m_c(x; C_k)\}_{c=1}^{n-1}$. In particular, our working model $L_c(x; \boldsymbol{\theta}_c^k)$ is

$$L_c(x; \boldsymbol{\theta}_c^k) = w_c^k \, m_c(x; C_k) + b_c^k, \qquad c = 1, \ldots, n-1, \quad (5)$$

where $\boldsymbol{\theta}_c^k = (b_c^k, w_c^k)$ are calibration parameters associated with class $c$ and the context size $k$. This affine structure directly targets the two primary sources of discrepancies between true and LLM logits: class-conditional shift and label marginal shift as illustrated in Equation (4). Specifically, by rearranging Equation (5) as $L_c(x; \boldsymbol{\theta}_c^k) = m_c(x; C_k) + [(w_c^k - 1)m_c(x; C_k) + b_c^k]$, we see that the term $(w_c^k - 1)m_c(x; C_k) + b_c^k$ serves as our learned approximation to the true correction term $\log(h(x, c, C_k)/h(x, 0, C_k))$. Within this learned correction, the intercept $b_c^k$ primarily addresses the query-independent "Label Marginal Shift" component from Equation (4), compensating for discrepancies in label priors. The query-dependent term $(w_c^k - 1)m_c(x; C_k)$ targets the "Class Conditional Shift" by allowing the slope $w_c^k$ to rescale the LLM's original logit $m_c(x; C_k)$.

Furthermore, $w_c^k$ enables the reorientation of the LLM's decision boundary. For instance, a negative $w_c^k$ inverts the LLM's initial assessment for a class relative to the reference, effectively correcting its predictive direction as illustrated in Figures 1 (c) and 4. This is a vital capability that methods merely learning a bias (i.e., fixing $w_c^k = 1$) lack. As detailed in Section 3.4, our framework not only unifies but also generalizes several recent ICL calibration techniques. Finally, it naturally encompasses the base LLM's original predictions as a special case when $b_c^k = 0$ and $w_c^k = 1$ for all $c$. In terms of learning the parameters, if an external calibration dataset $\{(x^{(j)}, y^{(j)})\}_{j=1}^{N_{cal}}$ is provided, we first compute the LLM's logits $\mathbf{m}(x^{(j)}; C_k)$ for each $x^{(j)}$. Then based on Equation (1), we estimate the parameters via minimizing the negative log-likelihood, i.e.,

$$\hat{\boldsymbol{\theta}}^k = \arg\min_{\boldsymbol{\theta}^k}\{\mathbb{L}_k(\boldsymbol{\theta}^k) \triangleq -\sum_{j=1}^{N_{cal}} \log f_{y^{(j)}}(\mathbf{m}(x^{(j)}; C_k); \boldsymbol{\theta}^k)\}, \tag{6}$$

where $\boldsymbol{\theta}^k = \{\boldsymbol{\theta}_c^k\}_{c=1}^{n-1}$ and $f_c(\mathbf{m}(x^{(j)}; C_k); \boldsymbol{\theta}^k) = \frac{\mathbf{1}_{\{c>0\}}\exp(L_c(x;\boldsymbol{\theta}_c^k)) + \mathbf{1}_{\{c=0\}}}{1 + \sum_{i=1}^{n-1}\exp(L_i(x;\boldsymbol{\theta}_i^k))}$. This optimization problem is equivalent to standard multi-class logistic regression using the model logits $m_c$ as input features. However, there is no external calibration dataset available beyond $C_k$. Therefore, we propose generating surrogate training data directly from the demonstration context $C_k$ via a leave-subset-out strategy. Specifically, we first select a context size $i$ such that $i < k$. We then construct the surrogate training dataset $\mathcal{T}_i$ using Algorithm 1 in Appendix E, as illustrated in Figure 2. Finally, we estimate calibration parameters $\hat{\boldsymbol{\theta}}^i$ via minimizing $\mathbb{L}_i$ under $\mathcal{T}_i$. Note that this method can be applied across multiple context sizes $i$, enabling ensembling extensions of $\{\hat{\boldsymbol{\theta}}^i\}_{i \in I}$ to construct a final estimator for calibration.

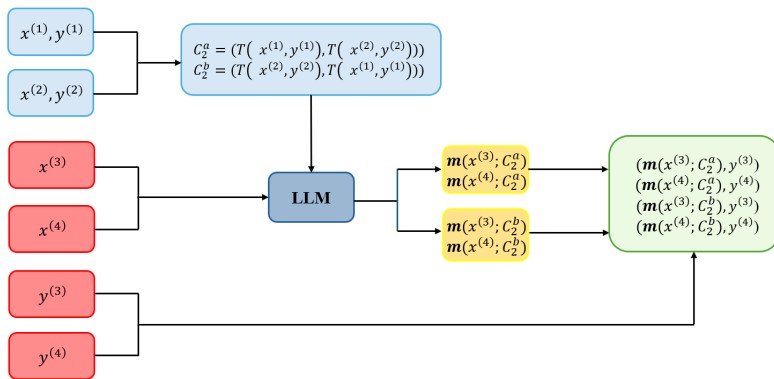

Figure 2: Surrogate data generation (Algorithm 1) for a 4-shot setting ($k = 4$) using a 2-shot context ($i = 2$). From the full set of 4 examples, many different 2-shot contexts (**blue**) can be formed; the figure illustrates two such possibilities. The remaining held-out examples (**red**) are used as queries with each context, and the LLM's logits are paired with the true labels to build a diverse surrogate dataset.

### 3.2.2 CONTEXT INVARIANCE AND DIRECTIONAL TRUST REGION

In the following subsection, we fix the context size $i \in I$ and introduce some enhancement on the proposed method. Note that our surrogate data generation process exposes a well-known limitation of ICL, its sensitivity to the composition and ordering of the context. Specifically, a single query pair $(x, y)$ is evaluated using multiple different sub-contexts $C_i$, yielding potentially different logits $\mathbf{m}(x; C_i)$ and label prediction for the same ground truth label $y$. In essence, an effective calibration method should mitigate this sensitivity, leading to more stable predictions. This motivates incorporating a mechanism to encourage context invariance in the calibrated predictions. To achieve this, we propose augmenting the standard MLE objective (Eq. (6)) with a context-invariance regularization term. Specifically, let $C_i^{(a)}$ and $C_i^{(b)}$ be any two distinct contexts of size $i$ drawn from $C_k$ for evaluating the same query $(x, y)$ in the surrogate data. We aim for the calibrated distributions $\boldsymbol{f}(\mathbf{m}(x^{(j)}; C_i^{(a)}); \boldsymbol{\theta}^i)$ and $\boldsymbol{f}(\mathbf{m}(x^{(j)}; C_i^{(b)}); \boldsymbol{\theta}^i)$, to be similar. To enforce this similarity, we utilize the symmetric cross-entropy between these two calibrated distributions as a regularizer defined as $L_{\mathrm{sym}}(\boldsymbol{\theta}^i, x, C_i^{(a)}, C_i^{(b)}) = \mathrm{H}\big(\boldsymbol{f}(\mathbf{m}(x^{(j)}; C_i^{(a)}); \boldsymbol{\theta}^i), \, \boldsymbol{f}(\mathbf{m}(x^{(j)}; C_i^{(b)}); \boldsymbol{\theta}^i)\big)$, where $\mathrm{H}(P, Q) \triangleq -\sum_{c=0}^{n-1}(P_c \log Q_c + Q_c \log P_c)$. This loss term measures the divergence between the two distributions induced by different contexts, penalizing differences in both directions. Then the overall penalty is defined by averaging $L_{\mathrm{sym}}$ over all possible pairs of contexts associated with each $x$.

$$\mathrm{InvPenalty}(\boldsymbol{\theta}^i) = \sum_{x} \sum_{\{C_i^{(a)}, C_i^{(b)}\}} L_{\mathrm{sym}}(\boldsymbol{\theta}^i, x, C_i^{(a)}, C_i^{(b)}). \tag{7}$$

The full expression of InvPenalty is given in Equation (14) of Appendix D. On top of ensuring context-invariance, a well-established calibration approach should also take into account the different scenarios induced by the base LLM's reliability and the size of the context. In particular, strong base LLMs warrant minimal adjustment, while weak ones require more aggressive correction, yet limited examples can mislead both cases, risking overfitting or under-correction. To balance this, we regularize the calibration by introducing a *directional trust region* that restricts parameter updates to remain aligned with the base LLM's logit. Specifically, we constrain the average cosine similarity between each parameter vector $\boldsymbol{\theta}_c^i = [b_c^i, w_c^i]^\top$ and the identity direction $v = [0, 1]^\top$, which corresponds to the base LLM via

$$\frac{1}{n-1} \sum_{c=1}^{n-1} \frac{(\boldsymbol{\theta}_c^i)^\top v}{\|\boldsymbol{\theta}_c^i\|_2} \geq \tau,$$

where $\|\cdot\|_2$ refers to $\ell_2$-norm and $\tau \in [0, 1]$ modulates the trust: large $\tau$ encourages minor scaling adjustments (exploitation), while smaller values permit broader corrections (exploration). This mirrors trust-region principles in policy optimization (e.g., TRPO (Schulman et al., 2015)), adapting model updates based on the confidence in prior predictions.

### 3.3 FULL ALGORITHM

The final optimization combines this constraint with the likelihood loss and a context-invariance regularizer:

$$\min_{\boldsymbol{\theta}^i} \left\{ \sum_{(\mathbf{m}^{(l)}, y^{(l)}) \in \mathcal{T}_i} -\log f_{y^{(l)}}(\mathbf{m}^{(l)}; \boldsymbol{\theta}^i) + \lambda_{\mathrm{inv}} \mathrm{InvPenalty}(\boldsymbol{\theta}^i) \right\} \text{ s.t. } \frac{1}{n-1} \sum_{c=1}^{n-1} \frac{(\boldsymbol{\theta}_c^i)^\top v}{\|\boldsymbol{\theta}_c^i\|_2} \geq \tau. \tag{8}$$

where $\lambda_{\mathrm{inv}} > 0$ is a hyperparameter controlling the strength of the context-invariance penalty. To solve this optimization problem, we used *SciPy's* `trust-constr` algorithm, a trust-region method designed for constrained optimization. This optimization can be carried out independently for each $i \in I \triangleq \{1, \cdots, k-1\}$, resulting in a set of calibration models $\{\hat{\boldsymbol{\theta}}^i\}_{i \in I}$, each specialized for a particular context length. Additionally,

at inference, any sub-context $C_i$ can be used to extract logits for a given size $i$. This paves the way for a *two-level ensembling strategy* to enhance robustness by aggregating predictions across both multiple context lengths and diverse sub-context samples. Specifically, we train multiple affine-logit models $\{\hat{\boldsymbol{\theta}}^i\}_{i \in I}$ using training sets with different sizes of the context. Then, at inference time, given a test query $x_{\text{test}}$, we first draw $\{C_i^{(j)}\}_{j \in \mathcal{M}_i}$ from $C_k$ for every $i \in I$, where $I$ and $\mathcal{M}_i$ are user-defined index sets with size $|\mathcal{M}_i|$ and $|I|$. Then we perform *intra-size* and *inter-size* ensembling by averaging the calibrated predictions over $\{C_i^{(j)}\}_{j \in \mathcal{M}_i}$ and across all context sizes $i \in I$ and output the predictive probability of SC for $x_{\text{test}}$ as

$$\hat{\mathbf{p}}_{\text{SC}}(x_{\text{test}}) = \frac{1}{|I|} \sum_{i \in I} \frac{1}{|\mathcal{M}_i|} \sum_{j \in \mathcal{M}_i} \boldsymbol{f}\big(\mathbf{m}(x_{\text{test}}; C_i^{(j)}); \hat{\boldsymbol{\theta}}^i\big). \tag{9}$$

The final predicted label is $\hat{y}_{\text{SC}} \in \arg\max_{y_c \in \mathcal{Y}} [\hat{\mathbf{p}}_{\text{SC}}]_c$. Overall, this ensembling procedure approximates marginalization over plausible sub-contexts and lengths, significantly improving calibration stability and accuracy. The full algorithm of SC is summarized in Table 2 of Appendix E.

### 3.4 Connections to Prior Work and Theoretical Insight

In this section, we show the connection of the proposed SC with the existing LM methods and provide a principle approach to theoretically understand these methods from the perspective of supervised learning. Specifically, LM methods rely on one core assumption.

**Assumption 1** *The correction term* $h(x, y, C_k) \propto \frac{1}{P_{LLM}(y|C_k)}$.

Under Assumption 1, the derivation in Section 3.2.1 yields that LM methods are equivalent to assuming

$$L_c^*(x) = m_c(x; C_k) + B_c(C_k), \qquad c = 1, \ldots, n-1, \tag{10}$$

where $B_c(C_k) = -\log[P_{\text{LLM}}(c|C_k)/P_{\text{LLM}}(0|C_k)]$. Therefore, they focus on optimally shifting the decision threshold of the base LLM via estimating $P_{\text{LLM}}(y|C_k)$, which thus gives an estimator for $B_c(C_k)$. We summarize the existing approaches of estimating $P_{\text{LLM}}(y|C_k)$ in Table 2 of Appendix D. However, Assumption 1 can be easily violated in practice, causing model mis-specification error. Therefore, instead of imposing Assumption 1, we propose to understand existing LM methods from the perspective of function approximation in the supervised learning. In this case, LM methods basically assume a working model (10). In contrast, the proposed SC considers a strictly larger working model:

$$L_c(x; \boldsymbol{\theta}_c^k) = w_c^k \, m_c(x; C_k) + b_c^k, \qquad c = 1, \ldots, n-1.$$

This offers a principle framework to compare SC with LM methods and indeed shows that SC generalizes existing LM methods. Furthermore, within this framework, we analyze these methods via statistical learning theory. Consider a dataset $\mathcal{T} = \{(x^{(j)}, y^{(j)})\}_{j=1}^N$ of size $N$, and denote by $\hat{f} := f_{\hat{\boldsymbol{\theta}}^k}$ the solution minimizing $\mathbb{L}_k(\boldsymbol{\theta}^k)$ under $\mathcal{T}$. Let $\mathcal{R}^*$ denote the Bayes risk and $\mathcal{R}(\hat{f})$ the 0-1 risk of $\hat{f}$. Then, under standard regularity conditions, the excess risk of SC satisfies, with high probability:

$$\underbrace{\mathcal{R}(\hat{f}) - \mathcal{R}^*}_{\text{excess risk}} \lesssim \underbrace{\sqrt{D_{\text{KL}}(P^* \,\|\, f^*) - D_{\text{KL}}(P^* \,\|\, P^*)}}_{\text{approximation error}} + \sqrt{\frac{2(n-1)}{N}}. \tag{11}$$

The decomposition leads to the following theoretical insight. Firstly, thanks to the strictly larger working model, SC attains an approximation error that is guaranteed to be no worse than that of LM methods. Secondly, SC estimates $2(n-1)$ parameters—one slope and one intercept per non-reference class—while LM methods estimate only $n-1$ parameters. This leads to a factor of 2 increase in estimation error, which scales with the number of parameters $d$ as $\mathcal{O}(d)$. This gives LM methods an advantage. However, SC incorporates several variance mitigation strategies to actively control estimation error and fully leverage its lower approximation error: (i) explicit regularization through the directional trust region constraint and context invariance penalty; and (ii) ensembling procedure in Algorithm 2.

# 4 EXPERIMENTS AND MAIN RESULTS

In this section, we validate the effectiveness of SC by evaluating its classification performance across three LLMs and nine benchmark datasets. SC consistently outperforms all baseline calibration methods across various settings, establishing a new state-of-the-art in ICL for classification.

## 4.1 EXPERIMENTAL SETUP

**Datasets.** We evaluate our method on nine text classification benchmarks covering sentiment, topic, and social media analysis: SST-2, SST-5 (Socher et al., 2013), AG News (Zhang et al., 2015), SUBJ (Wang and Manning, 2012), TREC (Li and Roth, 2002), Rotten Tomatoes (Pang and Lee, 2005), TweetEval-Emotion (Mohammad et al., 2018), TweetEval-Hate (Basile et al., 2019), and Financial PhraseBank (Malo et al., 2014).

**Models and Baselines.** We compare SC against the Base LLM and three prior calibration baselines (CC, BC, and DC) on three models: LLaMA-2-7B-Chat-HF (Touvron et al., 2023), Mistral-7B-Instruct-v0.3(Jiang et al., 2023), and Qwen2-7B-Instruct (Yang et al., 2024). All models are used off-the-shelf from Hugging Face without any fine-tuning. Appendix A provides full implementation details for the baselines.

**Evaluation.** Following prior work, we report Macro-F1 in 4-shot, 8-shot, and 16-shot settings. To ensure robustness, all results are averaged over 5 random seeds on a held-out test set of 256 examples per dataset. Our prompt template is described in Appendix C.

## 4.2 MAIN RESULTS

Figure 3 reports the Macro-F1 performance of five calibration methods across our full experimental suite (9 datasets, 3 LLMs, 5 seeds, and 3 few-shot settings). Notably, SC consistently achieves the highest score across all models and shot counts. In particular compared to the Base LLM, SC yields improvements of up to **+22.6%** absolute in Macro-F1 (8-shot on Qwen2-7B-Instruct), and on average provides **+11.1%** absolute gain across all models and shot configurations. Relative to the strongest competing calibration method (BC), SC further improves performance by up to **+13.4%** (16-shot on Mistral-7B-Instruct-v0.3) and achieves an average gain of **+7.1%**. Overall, these results confirm that SC offers a robust and generalizable enhancement of LM methods in few-shot learning. In addition, our numerical results are aligned with our theory in presented in Section 3.4. As shown in Figure 3, SC achieves the highest average score among all methods due to better approximation error, but also exhibits increased variance in its performance. More detailed numerical results and comparison are given in Appendix F. Furthermore, SC delivers a striking improvement on SST-5: in

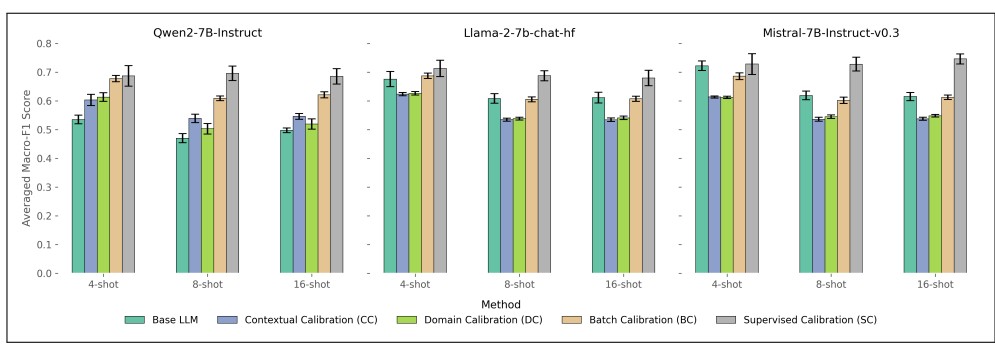

Figure 3: Average Macro-F1 scores for five methods across 9 datasets and 3 LLMs in 4-, 8-, 16-shots settings. Bars show the mean performance and standard deviation across datasets over 5 random seeds.

the 8-shot setting with Qwen, it boosts accuracy from 24% (base LLM) and 25% (other methods) to 44%, nearly doubling performance as shown in Figure 4. This substantial gain stems from SC's unique ability to not just shift logits, but to reverse the decision boundary when necessary as illustrated in Figure 1. For instance, it learns a bias of $-1.29$ and a weight of $-0.19$ for the *negative* class relative to *very negative*. This indicates that SC effectively shifts and reorients the LLM's decision boundary between closely related classes, enhancing overall performance.

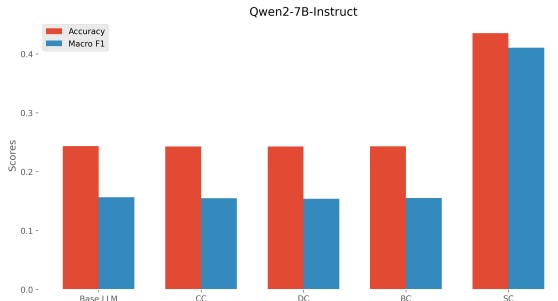

| Class | b | w |
|---|---|---|
| Very Negative (Ref.) | 0.000 | 0.000 |
| Negative | -1.294 | -0.188 |
| Neutral | 3.457 | 1.097 |
| Positive | 5.541 | 1.190 |
| Very Positive | -7.393 | 5.487 |

Figure 4: Performance on SST-5 with Qwen2-7B-Instruct in the 8-shot setting, averaged over 5 random seeds. The table on the right shows the average learned coefficients with respect to the *very negative* reference class.

**Ablations.** We conducted a series of ablation studies to validate the contributions of each component within our framework. First, we analyze the per-class scaling factor by comparing the full SC model against a variant, SC$^*$, that only learns the bias term (i.e., the scaling factor is fixed to 1). While SC$^*$ outperforms the baselines, which indicates estimating an optimal bias under SC framework is more effective than the methods employed by LM approaches, the full SC model performs even better. This confirms that learning to both shift and rescale logits is more advantageous. Second, we show that ensembling is highly effective: performance consistently improves as we aggregate calibrators trained on more different context sizes and average predictions over more sub-contexts at inference time. However, this performance gain comes at the cost of computational overhead, primarily at inference. The inference time scales linearly with the number of sampled sub-contexts, as each sample requires an additional forward pass. Furthermore, we confirm that both the directional trust-region constraint and the context invariance penalty are crucial and complementary components, with their combination yielding the highest performance. Finally, we validate that SC scales effectively to larger models, consistently delivering strong performance gains on a 13B parameter model across multiple datasets. Full results for the ablation studies are detailed in Appendix G.

## 5   CONCLUSION

In this paper, we introduce **Supervised Calibration (SC)**, a novel loss-minimization-based calibration framework designed to improve the performance of LLMs in ICL. We design SC to learn a class-specific affine transformation in logit space, allowing it to both shift and reorient the LLM's decision boundary. Thanks to its expressive functional form, we show that SC generalizes and extends the corrective capabilities of many existing calibration methods for ICL. Looking ahead, several avenues warrant exploration. First, performance could be improved by developing more principled approaches to context selection and weighting, moving beyond the current random sampling strategy. Second, a more rigorous theoretical analysis of SC is needed, particularly one that accounts for the statistical dependencies introduced by our surrogate data generation method. Finally, extending the principles of SC to calibrate LLMs for regression tasks presents a valuable direction for future research.

## REPRODUCIBILITY STATEMENT

We enable end to end reproducibility through: (i) an anonymized code repository with scripts to run Supervised Calibration (SC) and all baselines, linked via a main-text footnote ("Anonymized code for reproducibility," Page 3); (ii) complete algorithmic specifications in the paper, including the affine-logit model and leave-subset-out surrogate data (Section 3.2.1), the context-invariance and directional trust-region regularizers (Section 3.2.2), and the ensembling procedure (Section 3.3), with step by step pseudocode in Appendix E (Algorithms 1 and 2); (iii) an explicit statement of assumptions and theoretical insights in Section 3.4; (iv) full descriptions of datasets and model baselines in Section 4.1, and the exact prompt templates and label words in Table 1 of Appendix C; (v) a clearly defined evaluation protocol (Macro-F1, 4/8/16 shot settings, averaging over five random seeds on 256 held-out test examples) in Section 4.1; (vi) implementation and hyperparameter details in Appendix A, including compute resources, the invariance penalty weight ($\lambda_{\text{inv}}$), the schedule for $\tau$ in the trust region, and the number of sampled sub-contexts $m_i$; and (vii) comprehensive numerical results and ablations, including ensembling behavior and compute and timing, in Appendices F and G. Together, these materials are intended to support exact replication of all reported results.

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

## A    IMPLEMENTATION DETAILS

**Computation Resources.**    All large language models (LLMs) used in our experiments are based on publicly available implementations from the `Hugging Face Transformers` library (Wolf et al., 2020). We conduct all experiments on a dedicated computing node equipped with 8 NVIDIA A6000 Ada Generation GPUs.

**Contextual Calibration (Zhao et al., 2021)(CC)**    Following the original CC implementation, we compute the label probabilities conditioned on each of the three content-free tokens—'N/A', '', and '[MASK]'—along with the context. We then take the mean of these probabilities and use it to normalize the LLM's label-space probabilities computed for the test query and the same context.

**Domain-Context Calibration (Fei et al., 2023)**    We reproduce the DC baseline by using the test set as the unlabeled corpus to construct a bag-of-words. From this bag, we randomly sample tokens to create content-free and in-domain inputs with an average target length. This process is repeated 20 times, and we compute the mean probability over these samples. Following the original implementation, we use this mean to normalize the LLM's label-space probabilities computed for the test query and context.

**Batch Calibration (Zhou et al., 2023) (BC)**    BC is an inference-time calibration method that computes the mean of label probabilities over $m$ test samples given the context during the inference. We set $m = 128$ and use this mean to normalize the LLM's label-space probabilities given the test query and context.

**Supervised Calibration (SC)**    We adopt an ensembling strategy for SC as outlined in Algorithm 2. For each configuration—$k = 4$, $k = 8$, and $k = 16$—we set the minimum context size $i_{\min}$ (as defined in Algorithm 2) to 1, and the maximum context size $i_{\max}$ to $\min(5, k - 1)$. We fix the regularization parameter $\lambda_{\text{inv}}$ to 10 across all settings and LLMs. Additionally, the number of context to be sampled from $\mathcal{C}(i)$ (given in Definition 1) for size $i$ during the prediction is set as:

$$m_i = \min\left(\left\lfloor \frac{\mathcal{T}_i}{2} \right\rfloor, 24\right),$$

where $\mathcal{T}_i$ denotes the number of available samples for context size $i$.

To determine the value of $\tau$, we use the following formulation:

$$\tau = \arccos(\theta)$$

We first compute the in-sample accuracy of the LLM while generating the training data through Algorithm 1. Based on this accuracy, we set the value of $\theta$ as follows:

$$\theta = \begin{cases} 20^{\frac{1}{K-1}} & \text{if accuracy} \geq 0.9 \\ 45^{\frac{1}{K-1}} & \text{if } 0.7 \leq \text{accuracy} < 0.9 \\ 90^{\frac{1}{K-1}} & \text{if } 0.5 \leq \text{accuracy} < 0.7 \\ 180 & \text{if accuracy} < 0.5 \end{cases}$$

Here, $K$ denotes the number of distinct labels in the dataset.

While running SC with the setting $k = 4$, we excluded datasets containing more than four classes (i.e SST5 and TREC). This is because when the number of classes exceeds the number of context examples, some classes are inevitably left out of the training data. This imbalance poses a challenge for training logistic regression models across different context sizes.

## B ADDITIONAL RELATED WORK

**Calibration via centroids** . A parallel line of work mitigates in-context biases by replacing the standard decision rule with centroid-based classification. Han et al. (2022) proposed **Prototypical Calibration**, which models output probability vectors using Gaussian mixtures and assigns labels based on cluster likelihood, improving robustness to prompt variation and class imbalance. Similarly, Cho et al. (2024) introduced **Hidden Calibration**, which operates in the model's latent space by computing class centroids over hidden states and classifying based on proximity. Although these methods show empirical performance gains, they rely on additional data beyond the in-context examples, which may not always be available or compatible with the ICL setting.

**Mechanisms and prompt Optimization for ICL** Another line of work diagnoses why LLMs succeed or fail at ICL. The performance of a fixed prompt can swing from near random-guess to state of the art when the order of demonstrations is permuted (Lu et al., 2022), and it correlates strongly with the pre-training statistics of the tokens that appear in the prompt (Razeghi et al., 2022; Shin et al., 2022). From a theoretical perspective, ICL has been interpreted as implicit Bayesian inference in sequence models (Xie et al., 2022), while empirical evidence shows that sufficiently large models can even override entrenched semantic priors to learn arbitrary input–label mappings on the fly (Wei et al., 2023). A complementary literature focuses on controlling these factors. Template-search methods (Sørensen and Søgaard, 2022; Pan et al., 2023; Yin et al., 2023) and example-selection algorithms (Rubin et al., 2022; Liu et al., 2022b; Wan et al., 2023) systematically pick demonstrations that maximize mutual information or diversity, while Wan and colleagues (2023) add consistency and repetition checks. To make ICL more robust, researchers have proposed noisy–channel prompting (Seongjoo Min et al., 2022), flipped learning that trains the model against label noise (Ye et al., 2023), k-nearest-neighbour label assignment (Liangchen Xu et al., 2023), and lightweight decoder networks that adapt the prompt at inference time (Cui et al., 2023). Together, these studies paint a converging picture: effective ICL hinges on matching the prompt (template *and* examples) to the model's pre-training biases—then compensating for the remaining mismatches with task-specific selection or robust inference techniques.

## C PROMPT TEMPLATES

Table 1: Prompt templates and label words for various datasets.

| Dataset | Prompt Template | Label Words |
|---------|-----------------|-------------|
| SST2 | sentence: $<$x$>$\nsentiment: $<$y$>$ | negative, positive |
| SST5 | sentence: $<$x$>$\nsentiment: $<$y$>$ | terrible, bad, neutral, good, great |
| Rotten T. | review: $<$x$>$\nsentiment: $<$y$>$ | negative, positive |
| Financial P. | sentence: $<$x$>$\nsentiment: $<$y$>$ | negative, neutral, positive |
| Subj | review: $<$x$>$\ntype: $<$y$>$ | objective, subjective |
| TREC | question: $<$x$>$\ntarget: $<$y$>$ | abbreviation, entity, description, person, location, number |
| AGNews | news: $<$x$>$\ntopic: $<$y$>$ | world, sports, business, technology |
| TE-Emo | tweet: $<$x$>$\nemotion: $<$y$>$ | anger, joy, optimism, sadness |
| TE-Hate | tweet: $<$x$>$\nhate speech: $<$y$>$ | non-hate, hate |

## D  ADDITIONAL NOTATION AND DETAILED FORMULATION

Let $C_k = \{e^{(1)}, e^{(2)}, \ldots, e^{(k)}\}$ be the full demonstration set of $k$ unique input-label exemplars, where $e^{(l)} = (x^{(l)}, y^{(l)})$.

**Definition 1 (Set of Ordered Contexts)** *The set $\mathcal{C}(i)$ is defined as:*

$$\mathcal{C}(i) = \{(s_1, s_2, \ldots, s_i) \mid s_j \in C_k \quad for \quad j = 1, \ldots, i; \ and \ s_j \neq s_p \ for \ j \neq p\}. \tag{12}$$

*This set comprises all distinct ordered sequences (permutations) of $i$ unique exemplars chosen from the full demonstration set $C_k$.*

**Definition 2 (Set of Contexts Used for Query** $x$**)** *Given an exemplar $(x, y) \in C_k$, let $\mathcal{T}_i$ be the surrogate training dataset generated by Algorithm 1 using contexts of size $i$ from $C_k$. The set $\mathcal{C}(x, i)$ is defined as:*

$$\mathcal{C}(x, i) = \{C_i^{(j)} \in \mathcal{C}(i) \mid (x, y) \notin C_i^{(j)} \ and \ (m(x; C_i^{(j)}, y) \in \mathcal{T}_i\}. \tag{13}$$

*This set consists of all ordered contexts of size $i$ from $\mathcal{C}(i)$ that do not contain the specific exemplar $(x, y)$ itself, and were actually used to generate a (logit, label) pair for the query $x$ within the surrogate training data $\mathcal{T}_i$.*

**Definition 3 (Context Invariance Regularization Penalty)** *The total Context Invariance Regularization Penalty for parameters $\boldsymbol{\theta}^i$ is defined as:*

$$InvPenalty(\boldsymbol{\theta}^i) = \sum_{x \in \{x_l \mid (x^{(l)}, y^{(l)}) \in C_k\}} \sum_{\{C_i^{(a)}, C_i^{(b)}\} \subseteq \mathcal{C}(x,i), a \neq b} L_{sym}(\boldsymbol{\theta}^i, x, C_i^{(a)}, C_i^{(b)}). \tag{14}$$

*This penalty aggregates the symmetric cross-entropy loss over all distinct pairs of contexts $(C_i^{(a)}, C_i^{(b)})$ that were used to evaluate each unique query input $x$ derived from the original demonstration set $C_k$. It encourages the calibrated predictions for the same query $x$ to be consistent, regardless of the specific context $C_i^{(j)} \in \mathcal{C}(x, i)$ used to generate the intermediate LLM logits.*

Table 2: Summary of Label Based calibration methods. Each method adjusts the LLM prediction $P_{\text{LLM}}(y \mid x, C_k)$ via the different estimators of $P_{\text{LLM}}(y|C_k)$.

| Method | Formula | Description |
|--------|---------|-------------|
| LLM (Prob) | $\arg\max\limits_{y} P_{\text{LLM}}(y \mid x, C_k)$ | Selects the label with the highest conditional probability from the LLM. |
| Contextual Calibration (CC) | $\arg\max\limits_{y} \dfrac{P_{\text{LLM}}(y \mid x, C_k)}{P_{\text{LLM}}(y \mid \text{NA}, C_k)}$ | Normalizes the prediction using a content-free input to reduce label bias. |
| Domain-Context Calibration (DC) | $\arg\max\limits_{y} \dfrac{P_{\text{LLM}}(y \mid x, C_k)}{\frac{1}{N}\sum_i P_{\text{LLM}}(y \mid \text{RandDom}_i, C_k)}$ | Uses randomly sampled domain prompts as a reference for normalization. |
| Batch Calibration (BC) | $\arg\max\limits_{y} \dfrac{P_{\text{LLM}}(y \mid x, C_k)}{\frac{1}{N}\sum_i P_{\text{LLM}}(y \mid x_i, C_k)}$ | Calibrates by averaging predictions over a batch of reference inputs. |

# E  FULL ALGORITHMS

## Algorithm 1: Surrogate Data Generation for Calibration

---

**Require:** Demonstration set $C_k = \{(x^{(l)}, y^{(l)})\}_{l=1}^k$ of size $k$.
**Require:** Target context size $i$ such that $1 \le i < k$.
**Require:** LM inference function $\text{Infer}(x, C_i)$ that returns logit vector $\mathbf{m}(x; C_i)$.
 1: Initialize training set $\mathcal{T}_i \leftarrow \emptyset$.
 2: Generate $\mathcal{C}(i)$, the set of all distinct ordered subsets of $C_k$ with size $i$.  ▷ E.g., permutations of $C_k$, taking first $i$.
 3: **for** each context $C_i^{(a)} \in \mathcal{C}(i)$ **do**
 4:  Define the held-out set $R_i^{(a)} \leftarrow C_k \setminus C_i^{(a)}$.           ▷ Set difference based on elements.
 5:  **for** each query $(x, y)$ in $R_i^{(a)}$ **do**
 6:    Compute model logits vector: $\mathbf{m}(x; C_i^{(a)}) \leftarrow \text{Infer}(x, C_i^{(a)})$.
 7:    Add to training set: $\mathcal{T}_i \leftarrow \mathcal{T}_i \cup \{(m(x; C_i^{(a)}), y)\}$.    ▷ Store feature vector and true label.
 8:  **end for**
 9: **end for**
10: **Output:** Training set $\mathcal{T}_i$ consisting of pairs (model logits, true label).

---

## Algorithm 2: SC (Full Procedure)

---

**Require:** Full demonstration set $C_k = \{(x^{(l)}, y^{(l)})\}_{l=1}^k$; Set of context sizes $I = \{i_{min}, \ldots, i_{max}\}$; Regularization $\lambda_{inv} \ge 0$, $\tau \in [0, 1]$; Context samples $m_i \ge 1$; Query $x$; Inference function $\text{Infer}(x, C)$ returns logit vector $\mathbf{m}(x, C)$.
  **Part 1: Training Phase**
 1: Initialize parameter set $\Theta \leftarrow \emptyset$.
 2: **for** each context size $i \in I$ **do**
 3:  Generate training data $\mathcal{T}_i$ using Algorithm 1 with $C_k$.
 4:  Learn parameters $\hat{\boldsymbol{\theta}}^i$ by solving Eq. (8) using $\mathcal{T}_i, \lambda_{inv}, \tau$.
 5:  Store $\hat{\boldsymbol{\theta}}^i$ in $\Theta$.
 6: **end for**
  **Part 2: Prediction Phase (for query $x$)**
 7: Initialize list $P_{\text{list}} \leftarrow []$.
 8: **for** each context size $i \in I$ **do**
 9:  Sample index set $\mathcal{M}_i \subseteq \{1, \ldots, |\mathcal{C}(i)|\}$ uniformly at random such that $|\mathcal{M}_i| = m_i$.
10:  Retrieve learned parameters $\hat{\boldsymbol{\theta}}^i$ from $\Theta$.
11:  Retrieve sub-contexts $\{C_i^{(j)}\}_{j \in \mathcal{M}_i}$ from $\mathcal{C}(i)$ using $\mathcal{M}_i$.
12:  Initialize list $p_{\text{list}}^{(i)} \leftarrow []$.
13:  **for** $j \in \mathcal{M}_i$ **do**
14:    $\mathbf{m}(x, C_i^{(j)}) \leftarrow \text{Infer}(x, C_i^{(j)})$.
15:    $\mathbf{p}^{(j)}(x) \leftarrow \boldsymbol{f}(m(x, C_i^{(j)}); \hat{\boldsymbol{\theta}}^i)$.
16:    Append $\mathbf{p}^{(j)}(x)$ to $p_{\text{list}}^{(i)}$.
17:  **end for**
18:  $\hat{\mathbf{p}}_i(x) \leftarrow \frac{1}{m_i} \sum_{\mathbf{p}(x) \in p_{\text{list}}^{(i)}} \mathbf{p}(x)$.
19:  Append $\hat{\mathbf{p}}_i(x)$ to $P_{\text{list}}$.
20: **end for**
21: $\hat{\mathbf{p}}_{\text{SC}}(x) \leftarrow \frac{1}{|I|} \sum_{\mathbf{p}(x) \in P_{\text{list}}} \mathbf{p}(x)$.
22: **Output:** $\hat{y}_{\text{SC}} \in \arg\max_{y_c \in \mathcal{Y}} [\hat{\mathbf{p}}_{\text{SC}}(x)]_c$.

---

## F  DETAILED NUMERICAL RESULTS

In this section, we present detailed numerical results. For brevity, we refer to Qwen2-7B-Instruct, Llama-2-7b-chat-hf, and Mistral-7B-Instruct-v0.3 as Qwen, Llama, and Mistral, respectively, throughout the remainder of this section.

Table 3: Average Macro-F1 scores (%) for various calibration methods on selected datasets, evaluated for each LLM in the 4-shot setting ($k = 4$) over five random seeds. Values are presented as mean$_{s.d}$, with the highest score in each column highlighted in **bold** and shaded gray.

| Model | Method | Avg | AGNews | FPB | SST2 | RT | Subj | TE-Emo | TE-Hate |
|---|---|---|---|---|---|---|---|---|---|
| Qwen | Base LLM | 53.49 | $62.74_{1.56}$ | $31.22_{9.82}$ | $87.74_{7.42}$ | $88.23_{1.90}$ | $33.02_{0.81}$ | $35.23_{1.53}$ | $36.26_{0.20}$ |
| | CC | 60.30 | $85.22_{4.97}$ | $51.46_{10.52}$ | $\mathbf{91.63}_{0.78}$ | $89.91_{1.35}$ | $38.54_{7.64}$ | $35.07_{5.54}$ | $30.25_{0.00}$ |
| | DC | 61.30 | $\mathbf{88.68}_{0.68}$ | $52.86_{10.45}$ | $87.20_{5.76}$ | $\mathbf{90.31}_{0.90}$ | $36.97_{3.72}$ | $\mathbf{42.82}_{2.33}$ | $30.25_{0.00}$ |
| | BC | 67.71 | $70.14_{2.17}$ | $73.54_{2.75}$ | $88.92_{5.77}$ | $90.18_{1.41}$ | $\mathbf{74.10}_{3.92}$ | $40.94_{3.24}$ | $36.16_{0.00}$ |
| | SC | $\mathbf{68.66}$ | $72.76_{6.13}$ | $\mathbf{75.57}_{6.67}$ | $90.11_{4.99}$ | $89.39_{1.76}$ | $62.23_{11.15}$ | $41.25_{17.51}$ | $\mathbf{49.33}_{8.08}$ |
| Llama | Base LLM | 67.57 | $\mathbf{77.58}_{7.17}$ | $66.41_{5.92}$ | $93.36_{0.44}$ | $91.16_{1.59}$ | $40.18_{12.93}$ | $\mathbf{67.34}_{6.12}$ | $36.94_{7.64}$ |
| | CC | 62.31 | $71.01_{3.42}$ | $81.86_{2.72}$ | $93.17_{1.02}$ | $\mathbf{92.07}_{0.96}$ | $32.36_{0.00}$ | $35.45_{0.76}$ | $30.25_{0.00}$ |
| | DC | 62.61 | $72.10_{3.61}$ | $82.94_{2.82}$ | $93.60_{0.50}$ | $91.95_{1.18}$ | $32.36_{0.00}$ | $35.06_{1.02}$ | $30.25_{0.00}$ |
| | BC | 68.69 | $66.06_{2.04}$ | $\mathbf{84.56}_{3.75}$ | $93.53_{0.47}$ | $91.52_{1.28}$ | $54.15_{3.48}$ | $36.29_{1.38}$ | $\mathbf{51.70}_{2.00}$ |
| | SC | $\mathbf{71.28}$ | $71.76_{11.31}$ | $84.02_{4.70}$ | $\mathbf{94.25}_{0.53}$ | $91.56_{1.19}$ | $\mathbf{55.79}_{11.41}$ | $55.35_{10.57}$ | $46.20_{4.31}$ |
| Mistral | Base LLM | 72.20 | $\mathbf{79.28}_{6.90}$ | $89.55_{1.92}$ | $94.07_{0.75}$ | $92.47_{0.62}$ | $35.03_{6.42}$ | $\mathbf{60.53}_{9.67}$ | $54.51_{9.67}$ |
| | CC | 61.34 | $63.47_{1.91}$ | $87.24_{1.10}$ | $94.76_{0.70}$ | $92.39_{0.75}$ | $31.55_{0.00}$ | $32.11_{1.24}$ | $27.89_{0.00}$ |
| | DC | 61.17 | $63.29_{1.29}$ | $86.08_{2.53}$ | $94.17_{0.20}$ | $92.39_{0.75}$ | $31.55_{0.00}$ | $32.82_{1.37}$ | $27.89_{0.00}$ |
| | BC | 68.57 | $62.81_{1.11}$ | $86.66_{2.32}$ | $94.00_{0.69}$ | $\mathbf{92.63}_{0.67}$ | $48.05_{6.53}$ | $34.08_{2.67}$ | $\mathbf{61.73}_{2.67}$ |
| | SC | $\mathbf{72.78}$ | $75.66_{11.50}$ | $\mathbf{90.93}_{2.52}$ | $\mathbf{95.07}_{1.15}$ | $91.53_{2.51}$ | $\mathbf{59.38}_{12.89}$ | $59.48_{9.90}$ | $37.40_{16.36}$ |

Table 4: Average Macro-F1 scores (%) for various calibration methods on selected datasets, evaluated for each LLM in the 8-shot setting ($k = 8$) over five random seeds. Values are presented as mean$_{s.d}$, with the highest score in each column highlighted in **bold** and shaded gray.

| Model | Method | Avg | SST5 | TREC | AGNews | FPB | SST2 | RT | Subj | TE-Emo | TE-Hate |
|---|---|---|---|---|---|---|---|---|---|---|---|
| Qwen | Base LLM | 47.00 | $15.65_{0.33}$ | $45.40_{5.99}$ | $62.06_{0.79}$ | $30.13_{2.09}$ | $74.65_{18.64}$ | $91.00_{2.28}$ | $31.55_{0.00}$ | $34.55_{2.41}$ | $38.01_{0.00}$ |
| | CC | 53.91 | $15.48_{0.14}$ | $63.30_{5.09}$ | $82.27_{6.74}$ | $35.96_{7.09}$ | $89.00_{2.59}$ | $\mathbf{92.30_{1.37}}$ | $32.67_{0.96}$ | $46.29_{5.54}$ | $27.89_{0.00}$ |
| | DC | 50.26 | $15.41_{0.07}$ | $43.83_{3.18}$ | $\mathbf{86.86_{0.90}}$ | $35.92_{3.97}$ | $69.94_{19.04}$ | $91.09_{1.48}$ | $34.69_{4.03}$ | $46.74_{3.82}$ | $27.89_{0.00}$ |
| | BC | 60.88 | $15.52_{0.12}$ | $\mathbf{67.98_{1.73}}$ | $65.36_{1.18}$ | $66.87_{2.90}$ | $86.43_{4.45}$ | $91.95_{1.40}$ | $\mathbf{76.89_{1.32}}$ | $38.88_{3.00}$ | $38.01_{0.00}$ |
| | SC | $\mathbf{69.59}$ | $\mathbf{41.06_{2.80}}$ | $61.28_{4.30}$ | $85.32_{4.37}$ | $\mathbf{74.97_{6.19}}$ | $\mathbf{91.36_{3.75}}$ | $90.64_{2.56}$ | $70.94_{4.35}$ | $\mathbf{57.09_{19.29}}$ | $\mathbf{53.63_{3.26}}$ |
| Llama | Base LLM | 60.82 | $15.75_{1.31}$ | $\mathbf{44.60_{4.29}}$ | $74.55_{4.43}$ | $80.26_{2.73}$ | $94.15_{1.11}$ | $91.94_{1.17}$ | $37.54_{5.96}$ | $\mathbf{68.74_{3.60}}$ | $39.86_{8.28}$ |
| | CC | 53.44 | $30.61_{1.13}$ | $24.68_{2.68}$ | $64.66_{1.50}$ | $80.97_{2.81}$ | $94.59_{0.75}$ | $92.40_{0.72}$ | $31.55_{0.00}$ | $33.64_{1.28}$ | $27.89_{0.00}$ |
| | DC | 53.80 | $30.91_{1.25}$ | $25.52_{3.12}$ | $65.73_{0.68}$ | $82.44_{1.86}$ | $94.47_{1.29}$ | $92.47_{0.62}$ | $31.55_{0.00}$ | $33.25_{1.10}$ | $27.89_{0.00}$ |
| | BC | 60.52 | $23.49_{0.80}$ | $36.22_{1.47}$ | $63.78_{1.27}$ | $82.71_{3.05}$ | $94.09_{1.38}$ | $92.01_{1.03}$ | $\mathbf{65.21_{4.20}}$ | $33.56_{1.15}$ | $\mathbf{53.59_{2.51}}$ |
| | SC | $\mathbf{68.74}$ | $\mathbf{42.76_{4.23}}$ | $39.78_{10.65}$ | $\mathbf{86.01_{2.85}}$ | $\mathbf{85.58_{2.04}}$ | $\mathbf{95.27_{0.51}}$ | $\mathbf{92.53_{1.24}}$ | $61.89_{4.20}$ | $66.78_{5.65}$ | $48.05_{3.83}$ |
| Mistral | Base LLM | 61.86 | $14.66_{0.25}$ | $40.08_{5.39}$ | $70.59_{3.84}$ | $85.80_{4.22}$ | $94.41_{1.75}$ | $92.61_{0.45}$ | $37.20_{4.35}$ | $61.82_{3.01}$ | $59.55_{6.75}$ |
| | CC | 53.70 | $28.22_{1.26}$ | $27.80_{3.47}$ | $62.29_{1.42}$ | $84.64_{4.39}$ | $94.23_{1.79}$ | $92.69_{0.40}$ | $31.55_{0.00}$ | $32.95_{1.02}$ | $27.89_{0.00}$ |
| | DC | 54.47 | $31.15_{1.38}$ | $30.17_{3.26}$ | $62.07_{0.58}$ | $83.59_{3.07}$ | $\mathbf{94.68_{1.56}}$ | $\mathbf{92.70_{0.46}}$ | $31.55_{0.00}$ | $33.43_{0.80}$ | $27.89_{0.00}$ |
| | BC | 60.16 | $24.83_{0.54}$ | $40.26_{4.25}$ | $61.58_{0.97}$ | $83.59_{3.07}$ | $94.19_{1.52}$ | $92.62_{0.67}$ | $48.26_{7.71}$ | $32.91_{1.05}$ | $\mathbf{63.25_{2.06}}$ |
| | SC | $\mathbf{72.77}$ | $\mathbf{45.44_{3.01}}$ | $\mathbf{48.57_{8.36}}$ | $\mathbf{86.84_{3.42}}$ | $\mathbf{88.54_{4.70}}$ | $93.24_{1.58}$ | $90.09_{1.73}$ | $\mathbf{66.91_{6.13}}$ | $\mathbf{67.73_{7.99}}$ | $\mathbf{67.53_{11.74}}$ |

Table 5: Average Macro-F1 scores (%) for various calibration methods on selected datasets, evaluated for each LLM in the 16-shot setting ($k = 16$) over five random seeds. Values are presented as mean$_{s.d}$, with the highest score in each column highlighted in **bold** and shaded gray.

| Model | Method | Avg | SST5 | TREC | AGNews | FPB | SST2 | RT | Subj | TE-Emo | TE-Hate |
|---|---|---|---|---|---|---|---|---|---|---|---|
| Qwen | Base LLM | 49.75 | $14.47_{0.29}$ | $59.68_{5.52}$ | $63.10_{0.85}$ | $26.72_{0.84}$ | $87.55_{6.49}$ | $91.56_{1.80}$ | $31.55_{0.00}$ | $35.15_{0.56}$ | $38.01_{0.00}$ |
| | CC | 54.57 | $14.41_{0.21}$ | $69.40_{1.31}$ | $85.30_{2.77}$ | $27.16_{9.25}$ | $92.40_{0.89}$ | $93.32_{0.66}$ | $37.69_{4.80}$ | $43.58_{0.71}$ | $27.89_{0.00}$ |
| | DC | 51.92 | $14.38_{0.21}$ | $44.43_{3.81}$ | $\mathbf{88.07_{0.78}}$ | $39.48_{14.78}$ | $83.91_{9.82}$ | $\mathbf{93.42_{1.05}}$ | $35.32_{4.41}$ | $40.41_{1.50}$ | $27.89_{0.00}$ |
| | BC | 62.12 | $14.64_{0.36}$ | $\mathbf{72.75_{3.37}}$ | $69.02_{3.35}$ | $\mathbf{68.42_{8.43}}$ | $91.30_{0.91}$ | $92.64_{0.89}$ | $\mathbf{76.63_{3.03}}$ | $35.63_{0.92}$ | $38.01_{0.00}$ |
| | SC | $\mathbf{68.52}$ | $\mathbf{39.32_{6.66}}$ | $69.91_{2.56}$ | $85.34_{3.34}$ | $66.57_{9.62}$ | $\mathbf{92.95_{2.10}}$ | $92.15_{1.39}$ | $66.03_{10.62}$ | $\mathbf{53.63_{6.91}}$ | $\mathbf{50.76_{10.97}}$ |
| Llama | Base LLM | 60.72 | $14.49_{0.64}$ | $54.93_{5.18}$ | $75.64_{5.72}$ | $76.74_{5.43}$ | $94.25_{0.65}$ | $92.01_{1.17}$ | $37.00_{4.14}$ | $\mathbf{69.33_{9.55}}$ | $35.71_{2.64}$ |
| | CC | 53.42 | $31.40_{1.16}$ | $24.02_{4.02}$ | $63.73_{1.29}$ | $81.60_{2.58}$ | $94.41_{1.19}$ | $\mathbf{92.78_{0.67}}$ | $31.55_{0.00}$ | $33.37_{1.09}$ | $27.89_{0.00}$ |
| | DC | 54.06 | $32.09_{1.25}$ | $25.52_{3.12}$ | $65.54_{0.68}$ | $83.80_{3.50}$ | $\mathbf{94.59_{1.19}}$ | $92.47_{0.62}$ | $31.55_{0.00}$ | $32.35_{1.10}$ | $27.89_{0.00}$ |
| | BC | 60.72 | $24.61_{1.12}$ | $32.62_{3.83}$ | $63.85_{0.57}$ | $\mathbf{83.37_{3.68}}$ | $94.46_{0.85}$ | $92.46_{1.03}$ | $65.81_{2.42}$ | $33.64_{1.28}$ | $\mathbf{56.26_{4.20}}$ |
| | SC | $\mathbf{67.95}$ | $\mathbf{42.76_{4.23}}$ | $\mathbf{62.21_{5.62}}$ | $\mathbf{87.09_{2.82}}$ | $79.81_{8.37}$ | $93.81_{0.71}$ | $91.83_{1.46}$ | $50.65_{15.60}$ | $62.21_{4.15}$ | $46.72_{10.59}$ |
| Mistral | Base LLM | 61.49 | $14.42_{0.15}$ | $\mathbf{45.48_{4.45}}$ | $71.17_{2.31}$ | $84.17_{3.03}$ | $93.87_{0.79}$ | $92.39_{0.73}$ | $37.69_{3.27}$ | $\mathbf{70.79_{4.21}}$ | $43.42_{9.60}$ |
| | CC | 53.75 | $28.96_{1.12}$ | $28.97_{3.71}$ | $63.38_{0.91}$ | $82.73_{2.58}$ | $93.93_{0.35}$ | $\mathbf{92.93_{0.61}}$ | $31.55_{0.00}$ | $33.39_{1.09}$ | $27.89_{0.00}$ |
| | DC | 54.80 | $32.31_{0.33}$ | $32.79_{3.07}$ | $62.94_{0.85}$ | $85.17_{2.62}$ | $\mathbf{94.54_{0.80}}$ | $92.15_{0.49}$ | $31.55_{0.00}$ | $33.81_{1.16}$ | $27.89_{0.00}$ |
| | BC | 61.22 | $24.82_{1.23}$ | $41.11_{1.87}$ | $63.41_{0.84}$ | $81.51_{1.55}$ | $93.40_{0.58}$ | $92.46_{0.53}$ | $56.01_{6.53}$ | $33.64_{1.15}$ | $\mathbf{64.57_{0.98}}$ |
| | SC | $\mathbf{74.58}$ | $\mathbf{45.92_{3.25}}$ | $\mathbf{62.50_{3.97}}$ | $\mathbf{87.42_{1.83}}$ | $\mathbf{85.98_{4.47}}$ | $94.02_{1.88}$ | $91.07_{2.32}$ | $\mathbf{67.94_{10.40}}$ | $64.08_{4.31}$ | $\mathbf{72.34_{2.92}}$ |

Table 6: Average Accuracy scores (%) for various calibration methods on selected datasets, evaluated for each LLM in the 4-shot setting ($k = 4$) over five random seeds. Values are presented as mean$_{s.d}$, with the highest score in each column highlighted in **bold** and shaded gray.

| Model | Method | Avg | AGNews | FPB | SST2 | RT | Subj | TE-Emo | TE-Hate |
|---|---|---|---|---|---|---|---|---|---|
| Qwen | Base LLM | 68.01 | $75.23_{1.61}$ | $63.36_{2.91}$ | $87.93_{7.44}$ | $88.28_{1.85}$ | $48.16_{0.38}$ | $56.41_{1.52}$ | **$56.68_{0.08}$** |
| | CC | 64.34 | $85.47_{4.80}$ | $50.94_{13.11}$ | **$91.99_{0.67}$** | $89.92_{1.35}$ | $51.09_{4.15}$ | $37.58_{8.27}$ | $43.36_{0.00}$ |
| | DC | 65.87 | **$88.91_{0.58}$** | $52.73_{10.46}$ | $87.30_{5.81}$ | **$90.31_{0.90}$** | $50.04_{1.74}$ | $48.44_{3.54}$ | $43.36_{0.00}$ |
| | BC | **74.71** | $78.28_{1.53}$ | **$76.64_{2.85}$** | $89.06_{3.53}$ | $90.20_{1.41}$ | **$74.30_{3.84}$** | **$57.85_{1.53}$** | $56.64_{0.00}$ |
| | SC | 70.62 | $77.34_{3.89}$ | $74.69_{9.28}$ | $90.82_{4.17}$ | $89.41_{1.74}$ | $65.82_{8.12}$ | $45.08_{20.38}$ | $51.17_{6.97}$ |
| Llama | Base LLM | 72.86 | **$82.58_{4.17}$** | $78.55_{2.69}$ | $93.63_{0.40}$ | $91.17_{1.58}$ | $51.88_{7.02}$ | **$72.85_{5.23}$** | $46.33_{3.47}$ |
| | CC | 71.40 | $79.30_{2.02}$ | $85.51_{2.17}$ | $93.48_{0.94}$ | **$92.07_{0.95}$** | $47.85_{0.00}$ | $58.20_{0.92}$ | $43.36_{0.00}$ |
| | DC | 71.47 | $79.61_{1.89}$ | $85.94_{2.33}$ | $93.83_{1.19}$ | $91.95_{1.18}$ | $47.85_{0.00}$ | $57.77_{1.35}$ | $43.36_{0.00}$ |
| | BC | **74.05** | $77.19_{1.27}$ | **$86.99_{3.09}$** | $93.75_{0.48}$ | $91.52_{1.28}$ | **$58.20_{3.41}$** | $58.24_{1.68}$ | **$52.46_{1.97}$** |
| | SC | 73.78 | $78.12_{8.67}$ | $86.29_{2.88}$ | **$94.45_{0.47}$** | $91.56_{1.18}$ | $56.45_{10.80}$ | $61.05_{14.12}$ | $48.52_{1.90}$ |
| Mistral | Base LLM | **76.98** | $82.50_{4.17}$ | $90.47_{1.99}$ | $94.22_{0.76}$ | $92.50_{0.62}$ | $53.91_{0.00}$ | $68.36_{5.16}$ | **$56.88_{7.72}$** |
| | CC | 69.56 | $75.23_{2.33}$ | $87.34_{1.57}$ | $94.92_{0.70}$ | $92.42_{0.72}$ | $46.09_{0.00}$ | $52.27_{2.13}$ | $38.67_{0.00}$ |
| | DC | 69.44 | $75.31_{1.81}$ | $85.86_{3.40}$ | $94.38_{0.19}$ | $92.42_{0.72}$ | $46.09_{0.00}$ | $53.36_{2.16}$ | $38.67_{0.00}$ |
| | BC | 73.21 | $74.69_{1.57}$ | $87.19_{2.28}$ | $94.14_{0.70}$ | **$92.66_{0.67}$** | $48.44_{9.55}$ | $53.13_{1.40}$ | **$62.27_{2.35}$** |
| | SC | 75.59 | **$80.23_{9.04}$** | **$92.50_{1.87}$** | **$95.23_{1.09}$** | $91.56_{1.45}$ | **$62.03_{8.98}$** | $65.23_{14.12}$ | $42.27_{17.35}$ |

Table 7: Average Accuracy scores (%) for various calibration methods on selected datasets, evaluated for each LLM in the 8-shot setting ($k = 8$) over five random seeds. Values are presented as mean$_{s.d}$, with the highest score in each column highlighted in **bold** and shaded gray.

| Model | Method | Avg | SST5 | TREC | AGNews | FPB | SST2 | RT | Subj | TE-Emo | TE-Hate |
|---|---|---|---|---|---|---|---|---|---|---|---|
| Qwen | Base LLM | 60.32 | $24.34_{0.16}$ | $54.22_{7.21}$ | $73.16_{0.81}$ | $62.58_{0.52}$ | $76.09_{16.30}$ | $91.02_{2.26}$ | $46.09_{0.00}$ | $54.06_{2.74}$ | $61.33_{0.00}$ |
| | CC | 58.59 | $24.26_{0.08}$ | $67.34_{3.93}$ | $83.98_{4.99}$ | $33.28_{6.70}$ | $89.53_{2.29}$ | $92.30_{1.37}$ | $46.64_{0.47}$ | $51.33_{7.54}$ | $38.67_{0.00}$ |
| | DC | 55.94 | $24.26_{0.08}$ | $52.81_{1.70}$ | **$87.27_{0.79}$** | $33.13_{4.34}$ | $71.72_{16.75}$ | $91.09_{1.48}$ | $47.66_{2.05}$ | $56.88_{3.25}$ | $38.67_{0.00}$ |
| | BC | 68.64 | $24.30_{0.10}$ | **$73.59_{1.79}$** | $74.65_{0.69}$ | $70.70_{3.10}$ | $86.52_{4.48}$ | $91.95_{1.40}$ | **$77.03_{1.34}$** | $57.66_{1.63}$ | **$61.33_{0.00}$** |
| | SC | **72.30** | $43.52_{4.28}$ | $69.06_{1.32}$ | $86.02_{4.01}$ | $76.33_{6.70}$ | $91.88_{3.27}$ | $90.70_{2.46}$ | $72.50_{3.45}$ | $61.33_{20.50}$ | $59.38_{3.81}$ |
| Llama | Base LLM | 66.62 | $23.12_{0.67}$ | **$56.88_{4.73}$** | $80.08_{2.99}$ | $84.14_{3.44}$ | $94.30_{1.12}$ | $91.95_{1.17}$ | $48.75_{3.30}$ | **$75.39_{3.59}$** | $45.00_{5.04}$ |
| | CC | 63.59 | **$50.08_{3.00}$** | $36.48_{3.58}$ | $76.09_{1.24}$ | $82.73_{3.44}$ | $94.77_{0.72}$ | $92.42_{0.72}$ | $46.09_{0.00}$ | $55.00_{2.06}$ | $38.67_{0.00}$ |
| | DC | 63.71 | $48.59_{3.28}$ | $37.66_{3.71}$ | $76.80_{0.88}$ | $84.06_{2.90}$ | $94.61_{1.29}$ | **$92.50_{0.63}$** | $46.09_{0.00}$ | $54.37_{1.86}$ | $38.67_{0.00}$ |
| | BC | 66.55 | $31.48_{1.37}$ | $47.50_{1.69}$ | $75.78_{1.64}$ | $83.44_{3.69}$ | $94.22_{1.38}$ | $92.03_{1.04}$ | **$65.78_{4.30}$** | $54.77_{1.70}$ | **$53.91_{2.60}$** |
| | SC | **71.61** | $45.94_{5.52}$ | $50.86_{10.44}$ | **$86.56_{2.76}$** | $86.95_{2.35}$ | **$95.39_{0.52}$** | $92.58_{1.24}$ | $63.05_{3.39}$ | $73.52_{4.08}$ | $49.61_{3.55}$ |
| Mistral | Base LLM | 68.27 | $23.05_{0.25}$ | $51.48_{5.31}$ | $76.88_{1.66}$ | $85.86_{5.64}$ | $94.53_{1.75}$ | $92.66_{0.46}$ | $54.84_{1.88}$ | $74.30_{1.84}$ | $60.86_{5.49}$ |
| | CC | 64.42 | **$54.22_{0.52}$** | $40.86_{3.51}$ | $74.45_{1.67}$ | $84.61_{5.70}$ | $94.38_{1.76}$ | $92.73_{0.40}$ | $46.09_{0.00}$ | $53.75_{1.69}$ | $38.67_{0.00}$ |
| | DC | 65.02 | $54.06_{0.72}$ | $43.36_{3.44}$ | $74.30_{0.62}$ | **$86.64_{5.28}$** | $94.84_{1.51}$ | $92.73_{0.47}$ | $46.09_{0.00}$ | $54.45_{1.25}$ | $38.67_{0.00}$ |
| | BC | 66.35 | $34.92_{0.53}$ | $51.48_{4.13}$ | $73.67_{1.12}$ | $83.91_{4.18}$ | $94.30_{1.51}$ | $92.66_{0.67}$ | $48.75_{7.86}$ | $53.67_{1.76}$ | $63.83_{1.86}$ |
| | SC | **75.54** | $48.52_{5.84}$ | $57.58_{8.88}$ | **$87.42_{3.19}$** | **$89.53_{5.11}$** | $93.59_{1.41}$ | **$90.16_{1.68}$** | **$67.50_{6.08}$** | $75.16_{5.08}$ | **$70.39_{6.96}$** |

Table 8: Average Accuracy scores (%) for various calibration methods on selected datasets, evaluated for each LLM in the 16-shot setting ($k = 16$) over five random seeds. Values are presented as $\text{mean}_{s.d}$, with the highest score in each column highlighted in **bold** and shaded gray.

| Model | Method | Avg | SST5 | TREC | AGNews | FPB | SST2 | RT | Subj | TE-Emo | TE-Hate |
|---|---|---|---|---|---|---|---|---|---|---|---|
| | Base LLM | 62.71 | $22.81_{0.19}$ | $60.16_{3.27}$ | $75.62_{0.94}$ | $64.06_{0.00}$ | $87.66_{6.53}$ | $91.56_{1.81}$ | $46.09_{0.00}$ | $55.08_{0.96}$ | $\mathbf{61.33}_{0.00}$ |
| | CC | 59.08 | $22.81_{0.19}$ | $70.00_{1.29}$ | $86.17_{2.18}$ | $25.08_{8.06}$ | $92.66_{0.83}$ | $93.36_{0.65}$ | $49.30_{2.61}$ | $53.67_{1.99}$ | $38.67_{0.00}$ |
| Qwen | DC | 57.47 | $22.81_{0.19}$ | $51.02_{3.73}$ | $\mathbf{88.44}_{0.72}$ | $37.03_{15.89}$ | $84.14_{9.51}$ | $\mathbf{93.44}_{1.06}$ | $48.05_{2.37}$ | $53.59_{1.09}$ | $38.67_{0.00}$ |
| | BC | 69.49 | $22.89_{0.19}$ | $73.91_{2.07}$ | $78.05_{1.59}$ | $\mathbf{72.81}_{8.05}$ | $91.41_{0.92}$ | $92.66_{0.90}$ | $\mathbf{76.80}_{2.87}$ | $55.55_{0.62}$ | $61.33_{0.00}$ |
| | SC | **70.77** | $\mathbf{41.64}_{6.65}$ | $73.98_{2.67}$ | $85.78_{3.38}$ | $67.11_{11.83}$ | $\mathbf{93.20}_{1.89}$ | $92.19_{1.42}$ | $68.52_{8.27}$ | $\mathbf{60.23}_{5.91}$ | $54.30_{8.05}$ |
| | Base LLM | 66.97 | $22.50_{0.31}$ | $\mathbf{65.86}_{3.44}$ | $81.09_{2.71}$ | $81.72_{5.09}$ | $94.45_{0.57}$ | $92.03_{1.27}$ | $48.75_{2.09}$ | $\mathbf{73.20}_{6.24}$ | $43.12_{3.08}$ |
| | CC | 63.88 | $52.58_{0.80}$ | $37.19_{3.21}$ | $75.86_{1.47}$ | $82.34_{5.05}$ | $94.61_{1.09}$ | $\mathbf{92.81}_{0.68}$ | $46.09_{0.00}$ | $54.77_{0.76}$ | $38.67_{0.00}$ |
| Llama | DC | 64.11 | $50.70_{1.84}$ | $39.14_{4.17}$ | $76.95_{0.86}$ | $\mathbf{85.23}_{4.79}$ | $\mathbf{94.77}_{1.15}$ | $92.81_{0.80}$ | $46.09_{0.00}$ | $52.66_{1.70}$ | $38.67_{0.00}$ |
| | BC | 66.86 | $33.36_{1.32}$ | $44.22_{3.30}$ | $76.25_{0.80}$ | $83.52_{5.10}$ | $94.61_{0.83}$ | $92.34_{1.12}$ | $\mathbf{66.64}_{2.38}$ | $54.30_{0.35}$ | $\mathbf{56.48}_{4.29}$ |
| | SC | **70.92** | $44.45_{4.58}$ | $65.47_{4.60}$ | $\mathbf{87.42}_{2.90}$ | $78.83_{12.25}$ | $93.98_{0.77}$ | $91.88_{1.45}$ | $56.88_{8.83}$ | $66.64_{5.59}$ | $52.73_{11.41}$ |
| | Base LLM | 67.65 | $22.73_{0.16}$ | $56.88_{4.99}$ | $78.67_{1.92}$ | $83.91_{3.73}$ | $93.98_{0.80}$ | $92.42_{0.72}$ | $55.08_{1.44}$ | $\mathbf{76.48}_{3.62}$ | $48.67_{6.38}$ |
| | CC | 64.57 | $54.30_{0.55}$ | $42.66_{4.30}$ | $75.78_{1.05}$ | $82.03_{3.38}$ | $94.06_{0.38}$ | $\mathbf{92.97}_{0.61}$ | $46.09_{0.00}$ | $54.53_{1.74}$ | $38.67_{0.00}$ |
| Mistral | DC | 65.41 | $54.14_{0.72}$ | $47.11_{4.27}$ | $75.47_{1.03}$ | $85.08_{3.46}$ | $\mathbf{94.69}_{0.80}$ | $92.19_{0.49}$ | $46.09_{0.00}$ | $55.23_{1.76}$ | $38.67_{0.00}$ |
| | BC | 67.52 | $34.69_{1.32}$ | $53.28_{2.16}$ | $75.94_{1.01}$ | $81.25_{2.14}$ | $93.52_{0.58}$ | $92.50_{0.52}$ | $56.56_{6.32}$ | $55.00_{1.81}$ | $64.92_{0.90}$ |
| | SC | **76.96** | $47.27_{2.43}$ | $\mathbf{73.28}_{3.01}$ | $\mathbf{87.81}_{1.81}$ | $\mathbf{85.78}_{6.80}$ | $94.30_{1.70}$ | $91.09_{2.34}$ | $\mathbf{70.86}_{7.29}$ | $68.05_{5.05}$ | $\mathbf{74.22}_{1.38}$ |

# G ABLATION RESULTS

We conduct ablation studies to dissect the distinct contributions of key components within our Supervised Calibration (SC) framework.

## G.1 SCALING MATTERS

First, to isolate the impact of learning the per-class scaling factor $w_c$, which underpins SC's ability to reorient decision boundaries, we compare the full SC model against two alternatives in Figure 5: a restricted variant, SC* (where $w_c$ is fixed to 1, thus only learning an optimal bias term), and other baseline calibration methods. Our experiments reveal that SC* surpasses these other baselines. This suggests that estimating an optimal bias under SC framework is more effective than methods employed by LM methods. More critically, the full SC model achieves higher performance than SC*, suggesting that the flexibility to learn the scaling factor—and therefore to both shift and rescale the LLM's logits—offers a further advantage.

The performance difference between SC and SC* is particularly apparent on a challenging 8-shot, multi-class classification task (SST-5) where the base model's predictions are often poorly oriented. Specifically, Table 9 shows that SC* method achieves a very low Macro-F1 of 0.1004, indicating its inability to correct the model's predictions. In stark contrast, the full SC method boosts the Macro-F1 to 0.4106 and accuracy to 0.4352, representing a four-fold improvement. This vast performance gap confirms our hypothesis: on difficult tasks with severe miscalibrations, only full SC, capable of both shifting and scaling the decision boundary, can effectively correct severely misaligned LLM.

## G.2 ENSEMBLING ACROSS CONTEXT SIZES ($|I|$) IMPROVES PERFORMANCE

Second, we investigate whether ensembling calibrators trained with different context sizes improves predictive performance. Concretely, we train a collection of models $\{\hat{\boldsymbol{\theta}}^i\}_{i\in I}$, where each calibrator is fitted using training

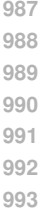

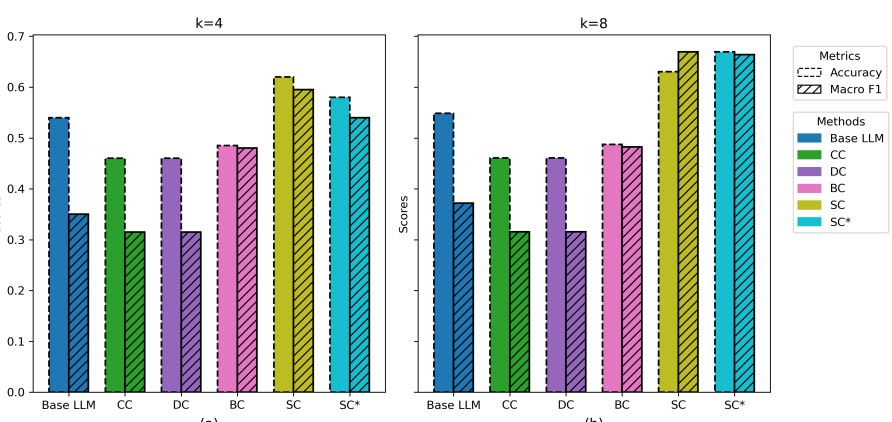

Figure 5: Accuracy and Macro-F1 scores of six methods on the Subjective dataset using the Mistral-7B-Instruct-v0.3 model in (a) 4-shot and (b) 8-shot settings. Results are averaged over 5 random seeds. Bars represent the mean performance for each metric as indicated in the legend. SC* stands for the case where the scaling factor $w_c$ is fixed to 1 under the SC framework. Notably, SC consistently outperforms all other methods in both settings. The improved performance of SC* over other baselines suggests that estimating an optimal bias under SC framework is more effective than the methods employed by LM approaches, while the full SC further demonstrates the advantage of also learning the scaling factor.

Table 9: Comparison on the 8-shot SST-5 task with the Qwen2-7B-Instruct model. SC v.s SC*.

| Method | Macro-F1 (mean $\pm$ SE) | Accuracy (mean $\pm$ SE) |
|---|---|---|
| Base LLM | $0.1565 \pm 0.0033$ | $0.2434 \pm 0.0016$ |
| SC* (scaling=1) | $0.1004 \pm 0.0125$ | $0.2227 \pm 0.0168$ |
| SC | $0.4106 \pm 0.0280$ | $0.4352 \pm 0.0428$ |

data with $i$ in-context examples. We then ensemble these context-size-specific calibrators and evaluate the impact of increasing the number of distinct $i$-shot learners in the ensemble (i.e., increasing $|I|$). Empirically, we observe a consistent and monotonic improvement in both Accuracy and Macro-F1 scores as $|I|$ grows as shown in Figure 6 and 7. This suggests that calibrators exposed to heterogeneous amounts of contextual information offer complementary signals, enhancing the robustness and predictive accuracy of the final calibrated output. These findings highlight a promising direction: with sufficient computational resources, one could train and ensemble an even broader set of context-specific calibrators to capture a richer diversity of contextual patterns, potentially unlocking further performance gains.

### G.3    MACRO-F1 GAINS AS THE NUMBER OF SAMPLED SUB-CONTEXTS INCREASES

Next, we investigate the impact of the number of sampled sub-contexts ($m_i$) used for prediction averaging within each context-size-specific calibrator during the ensembling phase. In Figure 8, our findings reveal that increasing $m_i$ (i.e averaging predictions over a greater number of distinct sub-contexts of size $i$) generally enhances Macro-F1 scores. This suggests that more comprehensive sampling of available context variations

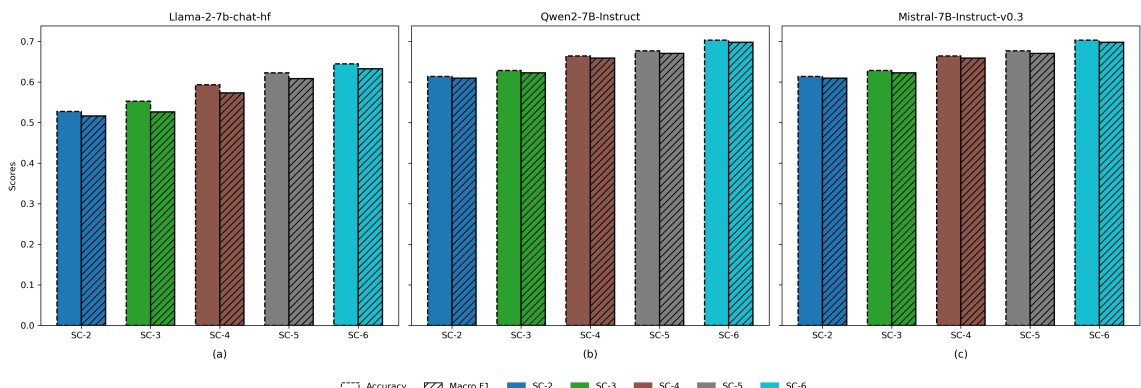

Figure 6: Impact of ensembling context-size-specific models within the SC framework on the Subjective dataset in an 8-shot setting. Results are reported for (a) Llama-2-7b-chat-hf, (b) Qwen2-7B-Instruct, and (c) Mistral-7B-Instruct-v0.3, using Accuracy and Macro-F1 scores averaged over 5 random seeds. Each ensemble, denoted SC-*N*, aggregates calibration models trained on context sizes ranging from 1 to $N$ (e.g., SC-2 uses models with context sizes 1 and 2, SC-6 includes context sizes 1 through 6). The consistent improvement in performance as N increases across all three LLMs highlights the general benefit of aggregating insights from a more diverse set of k-shot learners.

for each $i$-shot learner improves the accuracy of the ensemble's output, helping to further reduce ICL's sensitivity to specific context compositions.

### G.4 COMPUTE AND TIMING.

In Tables 10 and 11, we characterize the computational footprint of sub-context (SC) ensembling by reporting wall-clock training time $T_{\text{train}}$ and inference time $T_{\text{infer}}(m_i)$ per 256 test examples, where $m_i$ is the number of sampled sub-contexts with size $i$ used at inference for $SC_i$. Training is a one-time cost per method. SC rows are cumulative. Specifically, for $k = 4$ we aggregate $SC_2$–$SC_3$, and for $k = 8$ we aggregate $SC_2$–$SC_5$, whereas all bias-only baselines are effectively insensitive to $m_i$.

Specifically, SC ensembling increases inference time approximately linearly with $m_i$ because each additional sub-context entails an extra forward pass. This trend is evident at both context sizes. For $k = 4$, combining $SC_2$ and $SC_3$ adds a modest $T_{\text{train}} = 2.24$ s and yields $T_{\text{infer}}(1) = 22.91$ s, growing to $T_{\text{infer}}(6) = 134.96$ s, while baselines remain near 10.5 s regardless of $m_i$. For $k = 8$, the cumulative $SC_2$–$SC_5$ configuration requires $T_{\text{train}} = 489.62$ s and exhibits $T_{\text{infer}}(1) = 42.83$ s rising to $T_{\text{infer}}(6) = 260.32$ s, with baselines staying close to 11.1 s across all settings. These measurements are consistent with the simple cost model

$$T_{\text{infer}}(m_i) \approx m_i \times T_{\text{base},i} + \text{overhead},$$

in which $T_{\text{base}},i$ is the per-example cost of a single forward pass with context size $i$.

Practically speaking, When computation is a limiting factor, running the most effective single SC size offers a favorable accuracy–cost trade-off. In our experiments, $SC_3$ for $k = 4$ and $SC_5$ for $k = 8$ are the strongest individual calibrators, preserving most of the ensemble's accuracy gains while keeping inference overhead substantially closer to baseline runtimes.

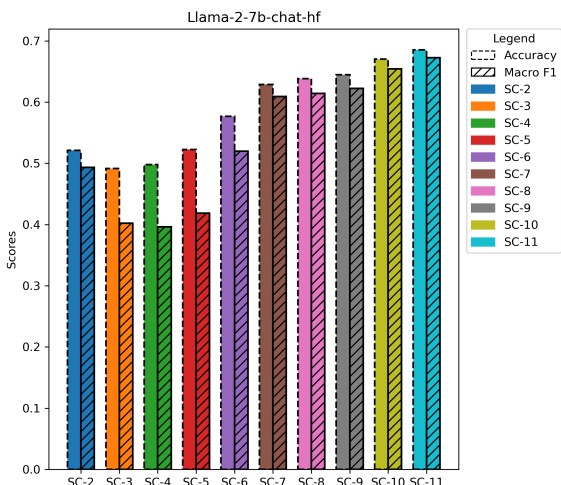

Figure 7: Impact of ensembling context-size-specific models within the SC framework on the Subjective dataset in an 16-shot setting. Result is reported for Llama-2-7b-chat-hf, using Accuracy and Macro-F1 scores averaged over 5 random seeds. Each ensemble, denoted SC-*N*, aggregates calibration models trained on context sizes ranging from 1 to $N$ (e.g., SC-2 uses models with context sizes 1 and 2, SC-11 includes context sizes 1 through 11). The consistent improvement in performance as N increases across all three LLMs highlights the general benefit of aggregating insights from a more diverse set of k-shot learners.

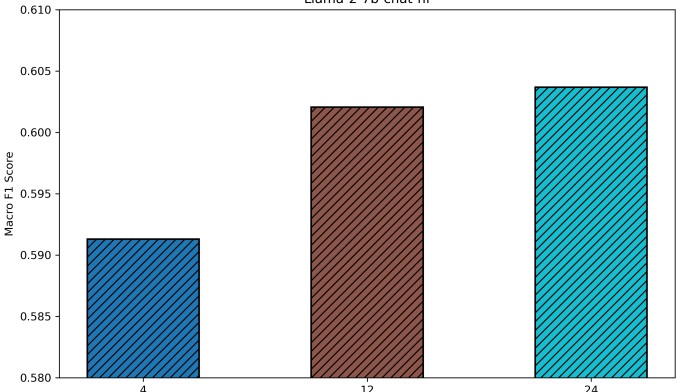

Figure 8: Impact of the number of sampled sub-contexts ($m_i$) used for prediction averaging within each context-size-specific model in the SC ensemble. Results show Macro-F1 scores on the Subjective dataset using the Llama-2-7b-chat-hf model in an 8-shot setting, averaged over 5 random seeds. The x-axis ($m_i$) represents the number of distinct contexts of a given size $i$ sampled to generate predictions, which are then averaged. Performance improves as more context variations are considered in the ensemble prediction.

Table 10: Training and inference timing (seconds) for $k = 4$.

| Method | $T_{train}(s)$ | $T_{infer}(1)$ | $T_{infer}(2)$ | $T_{infer}(3)$ | $T_{infer}(4)$ | $T_{infer}(5)$ | $T_{infer}(6)$ |
|---|---|---|---|---|---|---|---|
| Baseline | 0.00 | 10.51 | 10.51 | 10.51 | 10.51 | 10.51 | 10.51 |
| CC | 0.12 | 10.48 | 10.48 | 10.48 | 10.48 | 10.48 | 10.48 |
| Domain | 0.85 | 10.47 | 10.47 | 10.47 | 10.47 | 10.47 | 10.47 |
| Batch | 0.00 | 10.54 | 10.54 | 10.54 | 10.54 | 10.54 | 10.54 |
| $SC_2$ | 1.26 | 12.52 | 24.44 | 36.65 | 49.66 | 61.09 | 73.15 |
| $SC_3$ | 0.98 | 10.39 | 20.85 | 31.15 | 42.74 | 52.14 | 61.81 |
| **SC** | **2.24** | **22.91** | **45.29** | **67.80** | **92.40** | **113.23** | **134.96** |

Table 11: Training and inference timing (seconds) for $k = 8$.

| Method | $T_{train}(s)$ | $T_{infer}(1)$ | $T_{infer}(2)$ | $T_{infer}(3)$ | $T_{infer}(4)$ | $T_{infer}(5)$ | $T_{infer}(6)$ |
|---|---|---|---|---|---|---|---|
| Baseline | 0.00 | 11.34 | 11.34 | 11.34 | 11.34 | 11.34 | 11.34 |
| CC | 0.13 | 11.11 | 11.11 | 11.11 | 11.11 | 11.11 | 11.11 |
| Domain | 0.95 | 11.14 | 11.14 | 11.14 | 11.14 | 11.14 | 11.14 |
| Batch | 0.00 | 11.13 | 11.13 | 11.13 | 11.13 | 11.13 | 11.13 |
| $SC_2$ | 16.08 | 11.75 | 24.15 | 36.01 | 47.80 | 59.14 | 71.58 |
| $SC_3$ | 66.44 | 10.11 | 21.03 | 31.60 | 41.92 | 52.19 | 63.09 |
| $SC_4$ | 201.79 | 10.37 | 20.69 | 31.13 | 41.65 | 52.05 | 61.95 |
| $SC_5$ | 205.31 | 10.60 | 21.02 | 32.00 | 42.17 | 52.96 | 63.70 |
| **SC** | **489.62** | **42.83** | **86.89** | **130.74** | **173.54** | **216.34** | **260.32** |

## G.5 EFFECTS OF TRUST-REGION AND INVARIANCE

To isolate the impact of the key components of our proposed method, we conduct an ablation study, with the results presented in Table 12. We evaluate the performance contributions of our two main components: the directional trust-region constraint and the context invariance penalty.

The study begins with the "Uncalibrated (Baseline)" model, which achieves a Macro-F1 of 0.634. Introducing the core calibration mechanism without our proposed constraints ("No trust-region, no invariance") already yields a substantial improvement. When adding either the "Invariance only" or "Trust-region only" component, performance increases further, with both contributing similarly to the overall score. However, the full model, which combines both trust-region + invariance, achieves the highest performance across both Macro-F1 (0.746) and Accuracy (0.788). This demonstrates that both components are crucial and complementary, working together to deliver the best calibration results.

Table 12: Ablation study on the components of SC. Results show Macro-F1 and Accuracy, reported as mean $\pm$ standard error.

| Method | Macro-F1 $\pm$ SE | Accuracy $\pm$ SE |
|---|---|---|
| Uncalibrated (Baseline) | $0.634 \pm 0.008$ | $0.759 \pm 0.008$ |
| No trust-region, no invariance | $0.695 \pm 0.056$ | $0.729 \pm 0.047$ |
| Invariance only | $0.705 \pm 0.063$ | $0.741 \pm 0.054$ |
| Trust-region only | $0.706 \pm 0.060$ | $0.743 \pm 0.049$ |
| **Both: trust-region + invariance** | **$0.746 \pm 0.041$** | **$0.788 \pm 0.030$** |

### G.6 SCALING TO LARGER MODELS (LLAMA-13B)

To assess the scalability of our method, we ran additional experiments with the larger LLaMA-13B model. Due to computational constraints, we focused this scaling analysis on three datasets, **Rotten Tomatoes**, **SST-2**, and **AGNews**, where we compared its performance against the 7B variant. All experiments were conducted under the same 4-shot setup and averaged over 5 random seeds.

The results, presented in Tables 13, 14, and 15, demonstrate that our method, SC, scales effectively. Across all three datasets, SC consistently delivers the strongest performance on the LLaMA-13B model, achieving the highest Macro-F1 and Accuracy. Notably on AGNews, while the 7B baseline was competitive, SC provides a substantial improvement for the 13B model, boosting accuracy from 78.12 to 88.05. This confirms that our calibration approach remains highly effective and provides consistent benefits as the underlying language model size increases. We plan to incorporate further evaluations on even larger models in future work.

Table 13: Performance on the Rotten Tomatoes dataset with 7B and 13B models.

| Method | Macro-F1 (7B) $\pm$ SE | Accuracy (7B) $\pm$ SE | Macro-F1 (13B) $\pm$ SE | Accuracy (13B) $\pm$ SE |
|---|---|---|---|---|
| Baseline | $91.16 \pm 1.59$ | $91.17 \pm 1.58$ | $91.87 \pm 0.48$ | $91.89 \pm 0.49$ |
| CC | $\mathbf{92.06 \pm 0.96}$ | $\mathbf{92.07 \pm 0.95}$ | $92.33 \pm 0.11$ | $92.38 \pm 0.11$ |
| DC | $91.92 \pm 1.13$ | $91.95 \pm 1.18$ | $92.25 \pm 0.12$ | $92.29 \pm 0.10$ |
| Batch | $91.52 \pm 1.25$ | $91.52 \pm 1.28$ | $91.38 \pm 0.59$ | $91.41 \pm 0.57$ |
| SC | $91.56 \pm 1.19$ | $91.57 \pm 1.18$ | $\mathbf{92.33 \pm 0.26}$ | $\mathbf{92.38 \pm 0.25}$ |

Table 14: Performance on the SST-2 dataset with 7B and 13B models.

| Method | Macro-F1 (7B) $\pm$ SE | Accuracy (7B) $\pm$ SE | Macro-F1 (13B) $\pm$ SE | Accuracy (13B) $\pm$ SE |
|---|---|---|---|---|
| Baseline | $93.36 \pm 0.44$ | $93.63 \pm 0.40$ | $95.10 \pm 0.56$ | $95.21 \pm 0.56$ |
| CC | $93.17 \pm 1.92$ | $93.49 \pm 0.91$ | $94.81 \pm 0.74$ | $94.92 \pm 0.73$ |
| DC | $93.60 \pm 0.50$ | $93.83 \pm 1.19$ | $95.47 \pm 0.09$ | $95.61 \pm 0.10$ |
| Batch | $93.53 \pm 0.47$ | $93.75 \pm 0.48$ | $95.42 \pm 0.65$ | $95.51 \pm 0.65$ |
| SC | $\mathbf{94.25 \pm 0.53}$ | $\mathbf{94.45 \pm 0.47}$ | $\mathbf{95.65 \pm 0.26}$ | $\mathbf{95.80 \pm 0.25}$ |

Table 15: Performance on the AGNews dataset with 7B and 13B models.

| Method | Macro-F1 (7B) $\pm$ SE | Accuracy (7B) $\pm$ SE | Macro-F1 (13B) $\pm$ SE | Accuracy (13B) $\pm$ SE |
|---|---|---|---|---|
| Baseline | $\mathbf{77.58 \pm 7.17}$ | $\mathbf{82.58 \pm 4.17}$ | $85.74 \pm 1.77$ | $87.19 \pm 1.27$ |
| CC | $71.01 \pm 3.42$ | $79.30 \pm 2.02$ | $66.40 \pm 0.61$ | $77.73 \pm 0.28$ |
| DC | $72.10 \pm 3.61$ | $79.61 \pm 1.89$ | $66.90 \pm 1.00$ | $77.81 \pm 0.60$ |
| Batch | $66.06 \pm 2.94$ | $77.19 \pm 1.27$ | $66.32 \pm 0.63$ | $77.58 \pm 0.29$ |
| SC | $71.76 \pm 11.31$ | $78.12 \pm 8.67$ | $\mathbf{87.51 \pm 1.13}$ | $\mathbf{88.05 \pm 0.94}$ |

