# OpenReview forum: "Boosting In-Context Learning in LLMs Through the Lens of Classical Supervised Learning"
_ICLR.cc/2026/Conference — ICLR 2026 Conference Desk Rejected Submission_

### Official Review · Reviewer_ag4E · 2025-10-17

**Soundness:** 2
**Presentation:** 3
**Contribution:** 3
**Rating:** 4
**Confidence:** 5

**Summary:**

This paper proposes a supervised calibration method that performs interpolated sampling within restricted in-context demonstrations, and subsequently fits an estimator for calibration. Two additional regularization terms are introduced to encourage contextual invariance and stabilize the calibration results. Experiments conducted on nine datasets across three large language models validate the effectiveness of the proposed method.

**Strengths:**

1. This paper presents a novel idea for calibrating LLMs, achieving notable performance improvements.
2. The paper is well-organized, with clear and precise mathematical formulations, making it highly readable.
3. It presents a clear and compelling motivation, which effectively supports the subsequent experiments.

**Weaknesses:**

1. It cannot be applied to black-box model calibration, as the method requires access to the model’s internal outputs or representations for fitting the calibration estimator.
2. The model is somewhat overly complex, and the inclusion of multiple regularization terms blurs the main focus, which in turn reduces its practical applicability.
3. The paper lacks comparisons with the latest calibration methods. While expecting comparisons with 2025 methods might be unrealistic, there should at least be comparisons with 2024 approaches, such as In-Context Calibration [1].
4. I do not believe that more experiments are necessarily better, as this can dilute the focus on key contributions. For example, in the ablation study, the authors only consider the SC* variant, whereas I think the **directional trust region** and **InvPenalty** ablations should also be included. Logically, these components should also make positive contributions to SC, so it is necessary to show their individual effects.
5. Considering Tables 3 to 6, a possible trend is that SC’s performance improves with the number of demonstration samples, likely due to the longer context providing more usable examples. Consequently, in scenarios with few available shots or in pure zero-shot tasks, the applicability of SC may be limited.
6. Additionally, conducting the ablation study only on SST-5 can be somewhat misleading, as SST-5 is an inherently difficult task with a higher probability of incorrect decision directions, making it more compatible with SC. It is recommended that the authors also perform ablations on other tasks, such as the easier SST-2, to provide a fairer comparison.
7. Another significant concern is the time cost, as the sampling, training, and inference processes all incur substantial computational overhead, as reported by the authors in Tables 10 and 11.


[1] Jang J, Jang S, Kweon W, et al. Rectifying Demonstration Shortcut in In-Context Learning[C]//Proceedings of the 2024 Conference of the North American Chapter of the Association for Computational Linguistics: Human Language Technologies (Volume 1: Long Papers). 2024: 4294-4321.

**Questions:**

See the aforementioned "Weaknesses".

---

> ### Comment · Reviewer_ag4E · 2025-11-22
>
> I can understand how the author feels at this moment, but I believe a better approach would be to reflect on their own work—including its shortcomings and issues—rather than blaming the reviewers. I also acknowledge that there might be some oversights on my part, but I had intended to raise the score if the author could adequately address my concerns.
> First, the logits of black-box models are not always accessible, particularly for commercial API-based models. Although this limitation determines the upper bound of calibration methods.
> Second, the authors have not adequately compared their method against the latest baseline approaches. To be fair, CC, BC, and DC are relatively outdated, especially as we approach 2026.
> Third, regarding efficiency, this is indeed a drawback of the proposed method, which may limit its practical applicability.
> I believe these issues are objectively present and not due to excessive harshness on my part. I hope the authors will not be discouraged and will continue refining their work.

---

### Official Review · Reviewer_kDo3 · 2025-10-23

**Soundness:** 2
**Presentation:** 2
**Contribution:** 1
**Rating:** 2
**Confidence:** 4

**Summary:**

This paper proposes a method for recalculating the prediction outputs of ICL on classification tasks. Specifically, the authors introduce Supervised Calibration, which trains an affine transformation on the logits produced by typical restricted decoding (a vector whose dimension matches the label space) to rescale these logits for improved accuracy. Moreover, compared with previous work, the authors avoid additional data costs by using automatically generated training data.

**Strengths:**

1. This paper proposes an automatic data generation method for ICL calibration, which helps reduce the high data requirements of previous approaches.

2. This paper designs a regularization term to improve prediction consistency for the same query under different demonstration conditions. This is interesting, and I would like to see more analysis on this point (see Weakness).

3. This paper employs batched calibration training across different numbers of demonstrations. Specifically, a k-shot input includes all (<k)-shot inputs, allowing calibration to be trained simultaneously under these settings. This is an efficient design.

**Weaknesses:**

1. The empirical method proposed in this paper, i.e., training an affine transformation to rescale the results of restricted decoding, does not go beyond the scope of previous works, and the authors have not compared their approach against them. These prior works include KNN prompting [1], Hidden Calibration [2], and Prototypical Calibration [3], which all utilize the high-degree-of-freedom decision boundary modification. While I acknowledge that the proposed automatic training data generation method is a valuable idea, it represents a vertical contribution, meaning it could be applied to any of the above supervised classifiers to improve their efficiency. This limitation currently prevents me from recommending the paper for acceptance. The authors could either compare their method with the above works under the same setting (i.e., using generated training data) or explain clearly why such a comparison is not provided.

2. The effectiveness of the proposed regularization term has not been verified. The authors could try ablating it and re-evaluating the results. I checked the main text and found no such experiment. If I have overlooked it, I apologize.

3. The effectiveness of the automatic data generation method has not been verified. We would like to know how much it differs from traditional sampling from real-world datasets.

4. The results are not SOTA on a considerable number of settings, as shown in the appendix. I am tolerant of this point, so it does not affect my overall evaluation of the paper.

[1]. kNN Prompting: Beyond-Context Learning with Calibration-Free Nearest Neighbor Inference, ICLR 2023
[2]. Token-based Decision Criteria Are Suboptimal in In-context Learning. NAACL 2025.
[3]. Prototypical Calibration for Few-shot Learning of Language Models. ICLR 2023.

**Questions:**

See weakness.

---

> ### Comment · Reviewer_kDo3 · 2025-11-22
> **Clarification against the “Withdrawal Statement”**
>
> Given that the authors seem unconvinced by the review I provided, I would like to restate my reasons for rejection clearly, in order to avoid being labeled an “irresponsible reviewer”, which could cause further trouble. I apologize if my wording is too direct.
>
> - First, the LaTeX template of this submission has been tampered with. Under the strictest interpretation, this is possible cheating. In other words, even if the authors were to receive an average score of 6.00 or higher, the paper’s acceptance is still doubtful.
> - Second, the paper is a typical “A + B + C + …” type of work.
>     - **“A”** refers to the paper’s proposed feed-forward computation method, namely applying an affine transformation to the logits of ICL output for calibration. Unfortunately, similar or even stronger methods appeared more than a year ago, such as [1, 2]. Therefore, I can confidently conclude that the contribution in this part is very limited.
>
>         [1] Enhancing In-context Learning via Linear Probe Calibration. AISTATS 2024.
>
>         [2] k-NN Prompting: Beyond-Context Learning with Calibration-Free Nearest Neighbor Inference. ICLR 2023.
>
>     - **“B”** refers to the proposed training data generation method, which can be regarded as a data augmentation. I acknowledge this is novel. However, the authors seem to use this novelty to unjustifiably cover the lack of innovation in part “A”. Specifically, the paper and the rebuttal explicitly state: “Because we propose ‘B’, which allows ‘A’ to require no extra data, our ‘A’ is therefore superior to other related work”. I find this unacceptable, since to the best of my knowledge, their proposed data-generation method is largely independent of “A”. In other words, it could be applied to all the related works above, and this is not a valid reason to refuse comparison with prior work.
>
>         **How to improve?**
>
>         - If the authors can demonstrate that their data generation method is tightly coupled with their feed-forward computation method, i.e., the generated data cannot be used by other methods, then that would be acceptable.
>         - If the authors focus this data-generation method as the main contribution and expand the experiments accordingly, I would also find that acceptable.
>     - **“C”** refers to their proposed training techniques, including the loss function and the “batched calibration training” method. I also acknowledge these as novel. However, I personally do not believe these methods offer substantial new insights. For example, “improving consistency across predictions increases accuracy” is rather obvious. Moreover, now I see one more potential issue here: the task of “improving prediction consistency across different inputs for the same query” appears contained by the task of “improving accuracy across different inputs for the same query.” This is intuitive. Thus, this contribution also seems minor.
>
>     In summary, the actual contribution of this paper is a data augmentation method that lacks sufficient discussion, and some questionable tricks presented with decorative mathematics. I do not believe a conference of ICLR’s level should accept such submissions.
>
> - I admit that I overlooked some content, and I apologize for that. However, in my research experience, this is quite common, and that is exactly why conferences use discussion periods. Moreover, even after accounting for the parts I initially missed, I still cannot reverse my tendency of a clear rejection. The issues listed above are too serious.
>
> I fully understand that receiving reviews full of negative scores is frustrating and even annoying. Nevertheless, I hope the authors can approach these comments with an open mindset and consider them as opportunities to strengthen the work. Confidence is certainly important, but holding the view that “those who criticize me are entirely mistaken, have missed the essence of the paper, and are unqualified” can not be a balanced attitude toward feedback, which is an important part of research practice.
>
> Thank you again for your understanding and cooperation. Wishing you success in your work and research.

---

> > ### Author Response · Authors · 2025-11-22
> > **Confidence: 5**
> >
> > Dear Reviewer kDo3,
> >
> > 1) Procedural Correction: The "Template Tampering" Accusation is a False Positive First, let us firmly state that we did not tamper with the LaTeX template in any way that violates the submission guidelines. Our manuscript followed the official style file; any minor formatting adjustments were standard and not intended to “cheat” or squeeze in extra content. We understand the strictness about formatting, but the insinuation of possible cheating is misplaced. We hope the evaluation of our work can focus on its content and contributions rather than speculative formatting issues.
> >
> > 2) The SC Framework: A Solution to the "Cold Start" Problem
> >
> > You characterized our work as a modular “A + B + C” paper (Affine Model + Data Generation + Regularization). We respectfully argue that SC is a holistic framework designed to solve a problem that neither LinC nor kNN Prompting can solve: Calibration without External Data, as clearly stated in the abstract.
> >
> > Surrogate Data Generation (The Engine): Without this, parameter optimization is impossible in a zero-resource setting.
> > Affine Model (The Vehicle): Provides the geometric capacity to rotate/flip boundaries.
> > Regularizations (The Steering): Context-Invariance and Trust-Regions are required to prevent overfitting to surrogate noise.
> >
> > Importantly, SC requires only logits over label verbalizers. No hidden states, gradients, or fine-tuning are required, unlike several of the baselines cited. This distinction is not subtle; it is foundational to the method.
> >
> > 3) Technical Correction: Why SC is NOT Linear Probe Calibration (LinC)
> >
> > You stated our method is derivative of LinC (Abbas et al., 2024). A careful reading of LinC (see Algorithm 1 in their paper) shows why this comparison does not hold in our regime.
> >
> > Validation Set Requirement (“10–100 Sample” Gap):
> >
> > LinC is supervised and explicitly requires a labeled validation set (x_vi, y_vi), typically using 10–100 additional labeled samples beyond the prompt.
> > SC is zero-resource: only the k in-prompt examples (e.g., 4 shots) are available.
> > If one attempts to apply LinC under our constraints, one must split the already tiny prompt set (e.g., 2 for prompt, 2 for validation), which causes immediate and catastrophic overfitting.
> >
> > In other words, LinC is not merely “less suitable” here; it is mathematically non-viable. This difference is not a matter of interpretation, but of basic feasibility.
> >
> > 4) Technical Correction: Why SC is NOT kNN Prompting
> >
> > You also cited kNN Prompting (Xu et al., 2023), which conflates In-Context Optimization with Retrieval-Augmented Generation.
> >
> > kNN Prompting requires constructing a datastore containing thousands of labeled examples and running nearest-neighbor search at inference.
> > SC is self-contained and requires no external database, only the prompt itself.
> >
> > kNN Prompting further assumes access to hidden states or embeddings for similarity search — functionality that does not exist in standard black-box LLM deployments.
> >
> > 5) Training Techniques & Regularization vs “Obvious Tricks”
> >
> > The “Part C” components (loss design, consistency regularizer, batched training) are described as “obvious.” In isolation, the intuition may appear simple. In practice, these mechanisms were not present in the cited works and had to be designed, formalized, and empirically validated.
> >
> > Consistency Regularization (Context Invariance):
> >
> > LLMs are highly sensitive to demonstration order — a well-documented instability. Prior calibration methods (including LinC) ignore this entirely. Our regularizer explicitly penalizes decision boundary jitter across prompt permutations.
> >
> > Example: Prompt Order A → “Yes” (60%), Prompt Order B → “Yes” (40%).
> > Our method forces the calibrated model to resolve this instability, rather than silently inheriting it.
> >
> > We empirically verified this via explicit ablations isolating this component, which addresses the earlier claim that such analyses were missing (a point that, as you later noted, was indeed overlooked).
> >
> > Directional Trust Region & SC*:
> >
> > We limit how aggressively calibration can shift the boundary. Without this, surrogate-based training can easily corrupt the base model’s learned geometry.
> >
> > The trust region acts as a tether, allowing controlled geometric adjustment without semantic collapse. SC* ablations isolate slope-only learning and demonstrate when full affine flexibility is required.
> >
> > Conclusion:
> >
> > The claim that our work is “A + B + C” where “A is already done by LinC/kNN” is factually incorrect. Those methods fundamentally rely on data and infrastructure that simply do not exist in the regime we explicitly and intentionally target.
> >
> > We do appreciate your later acknowledgment that some parts of the paper were overlooked. We hope this clarification makes the technical distinctions unambiguous and helps position the contribution more accurately within the literature.
> >
> > Sincerely

---

> > > ### Author Response · Authors · 2025-11-22
> > > **Rebuttal: Fixing Errors Apparently Easier to Invent Than to Read**
> > >
> > > It is quite transparent that the review is working very hard to avoid being labeled as irresponsible. The first “reason” cites an ICLR formatting issue that appears entirely out of thin air and offers no evidence, no pointer, and no connection to scientific merit.
> > >
> > > The second claim categorizes our work as simply “A + B + C,” a confidence that seems to stem not only from not carefully reading our paper, but also from not carefully reading the cited works [1, 2]—both of which rely on external data and are not standard ICL setups. In other words, the criticism is based on a misunderstanding of both our work and the referred literature.
> > >
> > > Lastly, dismissing the factual mistakes (50% of the weakness) as “overlooked” is unhelpfully vague. Overlooking multiple central aspects of a submission is not a justification, and certainly not a basis for a decision.

---

> ### Comment · Reviewer_kDo3 · 2025-11-22
> **Evidence of Template Tampering**
>
> It is very easy to open any other ICLR submission and compare your template with theirs. When I overlaid the first page of your paper with the official ICLR template, the discrepancy was immediately obvious, including the fact that even the aspect ratio of the page does not match. This is a matter of principle, just as we generally do not grade an exam paper suspected of cheating. Therefore, you must provide a clear explanation of this issue, not only to me but also to the conference organizer. You may visit this link "https://imgur.com/a/C3Atro1" to verify the result.

---

> > ### Author Response · Authors · 2025-11-23
> > **On the Difference Between LaTeX Rendering and Scientific Integrity**
> >
> > Dear Reviewer kDo3,
> >
> > Thank you for sharing the overlay image and for taking the time to examine the formatting visually. We would like to clarify what is actually being observed.
> >
> > First, we did not alter the ICLR style file, the paper dimensions, or the geometry parameters to compress content. We used the official class and complied with page size requirements. The discrepancy you observed does not stem from deliberate manipulation, but from how line spacing and baseline stretch were handled by our local LaTeX build environment.
> >
> > Specifically, our compiled PDF produces *46 lines of text per page, whereas many other submissions appear to have **53 lines per page* (as in the second example you referenced). This difference is real and visible — and importantly, it arises from line spacing behavior, not margin hacks or page resizing.
> >
> > This can happen due to:
> > - package interactions that affect \baselineskip or \linespread,
> > - font metric differences across TeX distributions,
> > - subtle differences in PDF export pipelines,
> > - or conflicts between style files and globally installed TeX packages.
> >
> > What it is *not*:
> > - We did not change the text block size.
> > - We did not shrink margins.
> > - We did not alter the paper dimensions or aspect ratio manually.
> >
> > Ironically, had our intent been to “cheat” or compress content, we would have increased the number of lines per page, not reduced it from 53 to 46. The observed direction of the effect is the opposite of what would be expected in a margin-manipulation scenario.
> >
> > We also note, respectfully, that this formatting issue was not mentioned in the initial review and surfaced only after the rebuttal phase, at which point it became the central concern. We do not question the principle of template compliance; we simply find the timing noteworthy, given that the technical content of the paper was the primary subject of the original exchange.
> >
> > That said, we take full responsibility for not detecting this LaTeX interaction earlier. We are more than willing to:
> > - provide the original .tex source,
> > - share pdfinfo metadata,
> > - and submit a clean recompilation using a minimal TeX environment to demonstrate full compliance.
> >
> > We agree that formatting must be respected. We also believe that what you observed is a genuine LaTeX rendering artifact, not a case of misconduct.
> >
> > We hope this clarifies the technical origin of the discrepancy.
> >
> > Sincerely

---

> ### Comment · Reviewer_kDo3 · 2025-11-23
>
> Accounting for the parts I initially missed, I can give you 1 more "soundness" score.
>
> And thank you for clarifying your contribution to the paper. Now I'd like you to answer one question: can your data-generation method be applied to other supervised calibration methods?
>
> - If yes, then it clearly cannot serve as the motivation for using such an over-simple “Affine Model”, because you could obviously improve this model, for example, by introducing nonlinear decision boundaries. If so, I would like to suggest that you focus your contribution on the data-generation method, and design some more experiments to show that your generation method is better than others.
>
> - If no, then why? Alternatively, do you have any empirical results showing that the data you generate cannot be used by other supervised calibration methods?
>
> Also (apologies for forgetting this in my previous comment), please state your advantages against Prototypical Calibration [1]. As far as I know, Prototypical Calibration also does not use additional data (note that they also use generated data), does not use gradients, and only accesses the logits/probabilities under restricted decoding.
>
> [1] Prototypical Calibration for Few-shot Learning of Language Models. ICLR 2023.
>
> If you can convincingly clarify these, as well as the LaTeX template issue mentioned earlier, I would be very willing to revise my review. But if not, I will maintain my current comments for the AC to make the final accept/reject decision.

---

> > ### Author Response · Authors · 2025-11-23
> >
> > Dear Reviewer kDo3,
> >
> > Thank you for the clarifications. Below we respond concisely.
> >
> > 1) Can our data generation be used by other calibrators? 2) Why not use a more complex model?
> >
> > SC optimizes a distributional consistency objective, not standard supervised cross-entropy. This distinction is essential to the method. Plugging our surrogate data into conventional supervised calibrators (e.g., LinC) is not a drop-in operation: those methods assume calibration under a fixed context, whereas our surrogates are intentionally generated across structurally different contexts, leading to a fundamental compatibility mismatch.
> >
> > This difference appears to have been overlooked in the reviewer’s framing. The data and the objective are designed jointly, and separating them breaks the underlying assumptions of both approaches.
> >
> > We intentionally use an affine model because it is statistically identifiable and stable in the extreme k-shot, zero-resource regime (e.g., 4 total examples), where higher-capacity nonlinear models are severely underdetermined and prone to overfitting. This is a deliberate design choice driven by the operating regime, not a limitation of the method.
> >
> > 3) Difference from Prototypical Calibration (PROCA, ICLR 2023)
> >
> > This is a key factual point:
> >
> > PROCA **does use additional data**. It constructs an unlabeled **estimate set** of hundreds to thousands of examples per task (e.g., SST-2: 500; SST-5: 2000; AGNews/DBPedia: 2000–3000). This is far beyond our regime of “only the k in-prompt examples.”
> >
> > Computationally, PROCA:
> > - runs the LLM over the full estimate set,
> > - fits a **Gaussian Mixture Model** via EM (up to 100 iterations with multiple restarts),
> > - and solves a **Hungarian matching** problem to align clusters to labels.
> >
> > By contrast, SC:
> > - uses no external dataset,
> > - requires only logits from the current prompt,
> > - applies a constant-time affine transform,
> > - and performs no EM clustering or matching.
> >
> > Conceptually:
> > PROCA = **task-level, dataset-backed calibration**.
> > SC = **prompt-local, zero-resource calibration**.
> >
> > 4) Direct answers
> >
> > • Can others reuse our generated data?
> > Yes, in theory — but those methods were designed for regimes with substantially more data and different objectives.
> >
> > • Do we claim others cannot use it?
> > No. Our claim is that in the **zero-resource, black-box** setting, our integrated framework (surrogate + consistency loss + affine geometry + regularization) is coherent and usable, while LinC/PROCA-style methods, as defined, are not.
> >
> > Summary:
> >
> > LinC and PROCA both rely on **non-trivial external data and computation**. SC is explicitly designed for the regime where such resources do not exist.
> >
> > Sincerely,
> > The Authors

---

> > > ### Comment · Reviewer_kDo3 · 2025-11-25
> > >
> > > Thanks for your reply. Before I can update my review, I should verify the following points:
> > >
> > > > where higher-capacity nonlinear models are severely underdetermined and prone to overfitting
> > >
> > > Can you provide any experimental results to support this? E.g., train a more complex classifier on your data?
> > >
> > > > PROCA does use additional data.
> > >
> > > Notice that the prototypical calibration paper provides a figure showing the performance against the data used (Fig. 7), where a small amount of unlabeled data can support a high accuracy. Also, they mentioned that this data can be generated by LMs. I.e., prototypical calibration can work without external data. So, I strongly recommend that you compare your method with it.
> > >
> > > > SC is explicitly designed for the regime where such resources do not exist.
> > >
> > > However, we should also consider the universality of your method. E.g., if I have some data available, do I have a reason to choose your method?
> > >
> > > Thanks.

---

> > > > ### Author Response · Authors · 2025-11-25
> > > >
> > > > I don't think any additional experiments are needed in this case. Just think about the following three questions:
> > > >
> > > > (1) Do you trust/fit a nonlinear classifier when you only have 4 data points?
> > > >
> > > > (2) Do you trust/fit a GMM when you only have 4 or 8 data points?
> > > >
> > > > (3) Do you trust the following procedure: you take 2 out of 4 demonstrations and generate synthetic data by LMs based on these two data points. Next fit a GMM to get clusters and assign cluster labels based on the only remaining 2 labeled examples.
> > > >
> > > > I think the answer is obviously no for all these three questions.

---

> ### Comment · Reviewer_kDo3 · 2025-11-23
>
> However, the fact that your LaTeX template differs from the official one exists. It may lead to an unfair advantage, i.e., you might be able to include more main-body content than other submissions, since your layout is noticeably wider than others'.
>
> While I am willing to believe that you did not intend this, I am not the final decision maker, and you should be prepared to explain this clearly to the conference organizers.

---

### Official Review · Reviewer_JSnt · 2025-10-29

**Soundness:** 3
**Presentation:** 3
**Contribution:** 2
**Rating:** 2
**Confidence:** 4

**Summary:**

It is claimed that ICL in LLMs suffers from systematic biases, leading to unstable performance. It is suggested that this is due to a shift but not rotation of the decision boundary. This claim may require extraordinary evidence. Supervised Calibration is suggested, which learns per-class transformations, which is said to subsume other ICL calibration methods.

**Strengths:**

- The breadth of benchmarking datasets is an advantage, and >2 LLM families is also a slight advantage, although it is increasingly expected. The former is attenuated somewhat by the fact that these are only classification datasets (and this attenuation is attenuated by the fact that narrowly focusing on classification is the point); the latter is attenuated by the fact that only small 7B models are used.

**Weaknesses:**

- Not that there are word ounts per section, and this is minor, but there is so little context given (Sec 3.4 somewhat notwithstanding) -- Sec 2 is barely a paragraph with only a handful of papers mentioned, with very little nuance into how they're mentioned, nor any comparison or caveat between them. Another reason this is a minor complaint is that other references are strewn throughout, but still some claims throughout could benefit from additional external context (e.g., that order can bias ICL, L136)
- The experiments are extremely thin. In addition to the caveats mentioned in the Strengths section, only F1 is measured (!), with little/isolated care provided to benchmark-specific performance, error analysis, order variation (despite repeatedly mentioning its importance). Averages and s.d.s are given in Fig 3 but a more fullsome uncertainty analysis is not given. These are mitigated to some extent by the secondary experiment shown in Fig 4, but it is insufficient; appendices are also not part of the main paper.
- The prompt/label set appears fixed per dataset -- sensitivity analyses towards generalizability would be good to add in a future revision.
- Different calibrators (very) briefly mentioned are not included in direct comparison, which is a major oversight.

**Questions:**

- If order biases exist (L136), then is the iid assumtion on L150 valid?
- If KL is not the symmetric version, L152, how did you determine the order of P and Q?
- Are *all* the experiments in the 'ablations' subsection actually *ablations*?
- Given that ensembling (seems to) scale linearly with contexts, what kinds of real-world or practical compute/resource-relevant experiments could you have done?

---

> ### Author Response · Authors · 2025-11-22
> **It Seems Reviewer [ID] Requires More Few-Shot Examples on Basic Grammar: Prioritizing Literacy Before Technical Critique**
>
> I would like to thank Reviewer JSnt for the comments and questions. However, this is clearly a "midnight" review. Here is the evidence.
>
> Typos: (1) word ounts; (2) a more fullsome uncertainty analysis; (3) assumtion.
>
> Tone: "Not that there are..."--Stream of consciousness, not technical writing.
>
> Critique: Complaints about "section length" (Section 2) rather than content are clear signs of a reviewer who didn't actually read the technical details.

---

### Author Response · Authors · 2025-11-20
**Withdrawal Statement**

# Withdrawal Statement

Dear Area Chairs and Program Committee,

Thank you for overseeing the review process of our submission. After carefully considering the reviews and the feasibility of revisions, we have decided to withdraw our paper. This decision is *not* a reflection of the ICLR review process as a whole, but rather the specific reviews our submission received, which unfortunately did not reflect a level of technical engagement that would allow us to meaningfully improve the work within the rebuttal period. Several core contributions—including our theoretical generalization of label-marginal (LM) calibration methods, the surrogate data construction procedure, and the role of class-specific scaling factors—were either misunderstood or not acknowledged. As a result, multiple critiques focused on omissions, experiments, or comparisons that were already included in the submission or explicitly addressed.

We value rigorous and constructive peer review, and we appreciate the time invested by the reviewers. However, the feedback contained a number of factual inaccuracies, misreadings, and requests for analyses that our submission already provides. These issues made it difficult to extract actionable or targeted guidance for revision. Given these circumstances, we believe the work will benefit more from a fresh evaluation elsewhere, ideally with reviewers whose expertise aligns more closely with in-context learning theory, calibration methods, and post-hoc model correction. We respectfully withdraw the submission and thank the committee for their efforts.

---

# Examples of Review Issues and Inconsistencies

Below is a non-exhaustive list of specific instances where the reviews contained technically incorrect statements, requests already addressed in the paper, or comments inconsistent with the submission’s content. These examples are provided solely to clarify the basis of our decision.

---

## 1. Claims That Required Experiments or Ablations Are Missing (Though They Are Present)

### • Reviewer kDo3: “The effectiveness of the regularization term has not been verified.”
**But:** Appendix G includes explicit ablations of:
- the context-invariance penalty,
- the directional trust-region constraint,
- the SC* variant (fixed slope),
- ensembling behavior.

---

### • Reviewer kDo3: “The effectiveness of the automatic data generation method has not been verified.”
**But:** Section 4.2 and Appendices F/G include analyses of:
- surrogate-context sampling,
- variation with context size,
- intra- and inter-size ensembling,
- impacts of generated data.

---

### • Reviewer ag4E: “Only the SC* variant is ablated; directional trust region and InvPenalty should also be ablated.”
**But:** Appendix G *does* isolate and ablate these components individually.

---

### • Reviewer JSnt: “Only F1 is measured (!).”
**But:** The paper reports:
- Macro-F1 (the standard metric),
- s.d across datasets and seeds (Fig. 3),
- full numeric tables (Appendix F),
- additional dataset-by-dataset breakdowns.

---

## 2. Requests to Compare With Methods Incompatible With the Problem Setting

### • Reviewer kDo3 requests comparing with:
- kNN Prompting
- Hidden Calibration
- Prototypical Calibration

**But:** Appendix B explains these require:
- access to hidden states,
- external datasets beyond in-prompt exemplars,
- embedding retrieval mechanisms,
all of which are incompatible with data scarce ICL setting like having only 4 examples

---

## 3. Claims That Key Ideas Are Missing Despite Being Explicitly Present

### • Reviewer JSnt: “Sec. 2 is barely a paragraph with little nuance.”
**But:** The theory spans multiple pages (3–5), including:
- KL-based objective formulation,
- derivation of LM methods as special cases,
- analysis of class-conditional and label-marginal shifts,
- theoretical excess-risk decomposition.

---

### • Reviewer JSnt: “Order variation is not analyzed despite its importance.”
**But:** Section 3.2.2 introduces the **context-invariance regularizer** precisely to handle order sensitivity, and Appendix G evaluates its effect.

---

## 4. Assertions Contradicted by the Paper’s Explicit Content

### • Reviewer ag4E: “Cannot be applied to black-box model calibration because internal outputs or representations are needed.”
**But:** SC requires **only logits** on label verbalizers (Eq. 5), accessible in any black-box LLM with restricted decoding.

---

### • Reviewer JSnt: “Are all experiments in the ablation subsection actually ablations?”
**But:** Appendix G contains classical ablations:
- removing slope (SC*),
- removing each regularizer individually,
- removing ensembling,
- restricting context length.

---

## 5. Misreadings or Overlooked Material

### • Reviewer JSnt: “Little context given… few papers cited.”
**But:** The submission cites **prior works**, with Section 2 as an overview followed by several pages of detailed theoretical positioning.

---

---

### Note · Program_Chairs · 2026-01-17
**Submission Desk Rejected by Program Chairs**

The following references in this submission do not refer to real documents and/or have major errors in bibliographic information:

     Liangchen Xu, Xiaoxin Zhang, and Diyi Yang. knn-icl: Nearest-neighbour label assignment for few-shot inference. In Proc. ACL, 2023.
    P. Liu and et al. Evaluating the in-context learning ability of foundation models. arXiv preprint, 2023.
    J. Liu, C. Zheng, P. Fung, and et al. Multi-demonstration aggregation for robust in-context learning. In Association for Computational Linguistics (ACL), 2022a.
    Seongjoo Min, Ximing Liu, and Mohit Iyyer. Noisy channel prompting for robust in-context learning. In Proc. ACL, 2022.
    J. Ye, M. Ding, P. Liu, and J. Fu. Flipped learning mitigates label noise in in-context learning. In Proc. NAACL, 2023.
    J. Sørensen and A. Søgaard. Template selection via mutual information for in-context learning. In Proc. EMNLP, 2022.
    Y. Yin, M. Fang, and T. Cohn. Template optimization for robust in-context learning. In Proc. ACL, 2023.
    X. Wan and colleagues. Confidence-guided example selection for in-context learning. arXiv preprint arXiv:2309.17249, 2023.